# The FASTK family proteins fine-tune mitochondrial RNA processing

**Akira Ohkubo**[1], **Lindsey Van Haute**[2], **Danielle L. Rudler**[3,4,5], **Maike Stentenbach**[3,4,5], **Florian A. Steiner**[6], **Oliver Rackham**[3,4,7,8,9], **Michal Minczuk**[2], **Aleksandra Filipovska**[3,4,5,9,10‡]*, **Jean-Claude Martinou**[1‡]*

**1** Department of Cell Biology, University of Geneva, Geneva, Switzerland, **2** Medical Research Council Mitochondrial Biology Unit, University of Cambridge, Cambridge, United Kingdom, **3** Harry Perkins Institute of Medical Research, Queen Elizabeth II Medical Centre, Perth, Australia, **4** ARC Centre of Excellence in Synthetic Biology, Queen Elizabeth II Medical Centre, Perth, Australia, **5** Centre for Medical Research, The University of Western Australia, Queen Elizabeth II Medical Centre, Perth, Australia, **6** Department of Molecular Biology, University of Geneva, Geneva, Switzerland, **7** School of Pharmacy and Biomedical Sciences, Curtin University, Perth, Australia, **8** Curtin Health Innovation Research Institute, Curtin University, Perth, Australia, **9** Telethon Kids Institute, Perth Children's Hospital, Perth, Australia, **10** School of Molecular Sciences, The University of Western Australia, Perth, Australia

‡ The authors wish it to be known that, in their opinion, the last two authors should be regarded as joint Last Authors.
* aleksandra.filipovska@uwa.edu.au (AF); Jean-Claude.Martinou@unige.ch (J-CM)

**Data Availability Statement:** RNA-seq datasets have been deposited in the NCBI GEO with accession numbers GSE156556 (HITS-CLIP) and GSE156260 (mitochondrial transcriptomics). The

## Abstract

Transcription of the human mitochondrial genome and correct processing of the two long polycistronic transcripts are crucial for oxidative phosphorylation. According to the tRNA punctuation model, nucleolytic processing of these large precursor transcripts occurs mainly through the excision of the tRNAs that flank most rRNAs and mRNAs. However, some mRNAs are not punctuated by tRNAs, and it remains largely unknown how these non-canonical junctions are resolved. The FASTK family proteins are emerging as key players in non-canonical RNA processing. Here, we have generated human cell lines carrying single or combined knockouts of several FASTK family members to investigate their roles in non-canonical RNA processing. The most striking phenotypes were obtained with loss of *FASTKD4* and *FASTKD5* and with their combined double knockout. Comprehensive mitochondrial transcriptome analyses of these cell lines revealed a defect in processing at several canonical and non-canonical RNA junctions, accompanied by an increase in specific antisense transcripts. Loss of *FASTKD5* led to the most severe phenotype with marked defects in mitochondrial translation of key components of the electron transport chain complexes and in oxidative phosphorylation. We reveal that the FASTK protein family members are crucial regulators of non-canonical junction and non-coding mitochondrial RNA processing.

## Author summary

As a legacy of their bacterial origin, mitochondria have retained their own genome with a unique gene expression system. All mitochondrially encoded proteins are essential

mass spectrometry proteomics data have been deposited to the ProteomeXchange Consortium via the PRIDE partner repository with the dataset identifier PXD020993 and 10.6019/PXD020993. All other relevant data are within the manuscript and Supporting information.

**Funding:** This work was supported by fellowships and project grants from the Swiss National Science Foundation (31003A_179421 to JCM and 31003A_175606 to FAS), the National Health and Medical Research Council (insert APP1154646, APP1154932, APP1159594 and APP1156747 to AF and OR), the Australian Research Council (DP170103000 to AF and OR), the Cancer Council of Western Australia (to AF), and Medical Research Council, UK (MC_UU_00015/4 to M.M.). DR and MS are supported by UWA Postgraduate Scholarships. The funders had no role in study design, data collection and analysis, decision to publish, or preparation of the manuscript.

**Competing interests:** The authors have declared that no competing interests exist.

components of the respiratory chain. Therefore, the mitochondrial gene expression is crucial for their iconic role as the 'powerhouse of the cell'–ATP synthesis through oxidative phosphorylation. Consistently, defects in enzymes involved in this gene expression system are a common source of incurable inherited metabolic disorders, called mitochondrial diseases. The human mitochondrial transcription generates long polycistronic transcripts that carry information for multiple genes, so that the expression level of each gene is mainly regulated through post-transcriptional events. The polycistronic transcript first undergoes RNA processing, where individual mRNA, rRNA, and tRNA are cleaved off. However, its entire molecular mechanism remains unclear, and in particular, 'non-canonical' RNA processing has been poorly understood. To address this question, we studied the FASTK family proteins, emerging key mitochondrial post-transcriptional regulators. We generated different human cell lines carrying single or combined disruption of *FASTKD3*, *FASTKD4*, and *FASTKD5* genes, and analyzed them using biochemical and genetic approaches. We show that the FASTK family members fine-tune the processing of both 'canonical' and 'non-canonical' mitochondrial RNA junctions.

## Introduction

The circular 16.5 kbp human mitochondrial DNA (mtDNA) encodes 37 genes including two rRNAs, 22 tRNAs, and 13 ORFs, all of which are essential for oxidative phosphorylation (OXPHOS) [1]. Mitochondrial transcription initiates at two distinct divergent promoters, one on each strand, leading to two almost genome-length polycistronic transcripts: the primary transcript generated from the heavy strand promoter (HSP) includes 8 monocistronic mRNAs, two bicistronic mRNAs (*ND4L/4* and *ATP8/6*), 14 tRNAs, and two rRNAs, while the complementary primary transcript derived from the light strand promoter (LSP) encodes only one mRNA (*ND6*), 8 tRNAs, and long stretches of non-coding sequences [2,3]. As a consequence of the polycistronic organization of the primary transcripts, the expression level of each individual gene is mainly controlled through post-transcriptional events, including stabilization, processing and modification [4–6]. Most mRNAs and rRNAs within the polycistronic transcripts are flanked by tRNAs, and cleavage at the tRNA junctions by RNase P (MRPP1-3) and RNase Z (ELAC2) allows excision of the individual transcripts [7–13]. This is referred to as the 'tRNA punctuation model' [14,15], and the flanking tRNAs are named canonical processing junctions. However, this model does not explain the processing of mRNAs that lack flanking tRNAs. In human mitochondria, junctions not flanked by tRNA, here named non-canonical junctions, are found at the 3′ UTR of *ND6*, 5′ UTR of *CO1*, between *ND5* and *CYTB*, and between *ATP8/6* and *CO3* mRNAs. The machinery responsible for processing these non-canonical junctions is currently unknown. Recent work has identified several candidates involved in the process, including the Fas-activated serine/threonine kinase (FASTK) family proteins [11,16–19].

The FASTK family includes six RNA binding proteins (RBPs), FASTK and FASTKD1-5 [20–23], all of which are localized in the mitochondrial matrix (Fig 1A) [16,18–20,24–27]. The available evidence to date suggests that the FASTK family proteins are broadly involved in post-transcriptional regulation, ranging from RNA maturation to translation, although the precise mode of action of different members of the family remains to be elucidated [20].

FASTKD2 was described as part of a protein complex together with RPUSD3, RPUSD4, NGRN, WBSCR16 and PTCD1, involved in pseudouridylation of the *16S* rRNA [19,28–30]. Mutations in the *FASTKD2* gene have been identified in patients with a MELAS

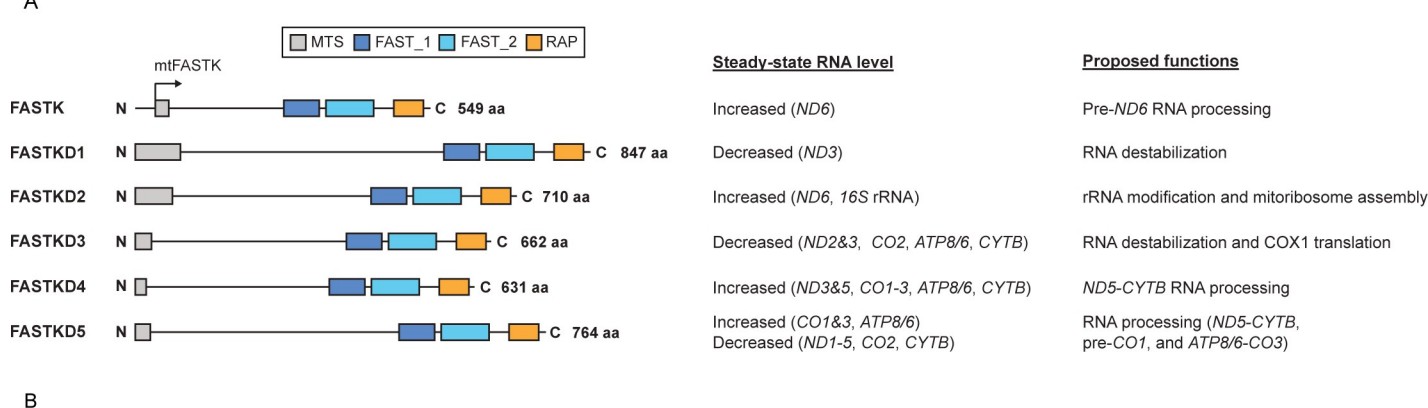

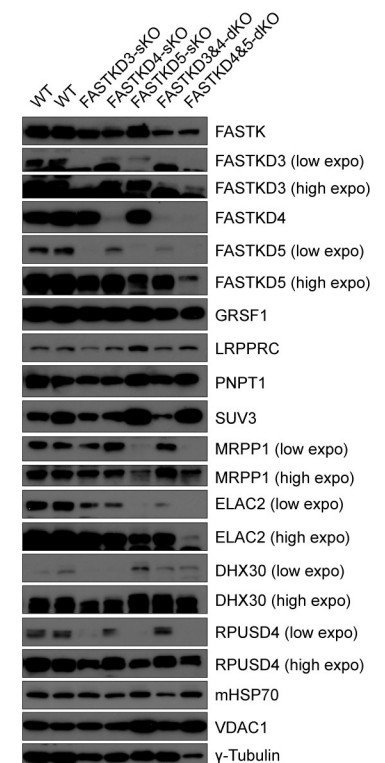

**Fig 1. Generation of HAP1 cell lines carrying single or combined knockouts of *FASTK* family members.** (**A**) Overview of the FASTK protein family and their proposed roles in mitochondrial post-transcriptional regulation. All proteins share a conserved C-terminal domain architecture of FAST_1, FAST_2, and the RAP domains. An internal translation start site exposes a mitochondrial targeting sequence (MTS) leading to expression of the mitochondrial FASTK (mtFASTK). (**B**) Expression levels of FASTK family proteins and mtRBPs involved in the post-transcriptional regulation in WT (control) and *FASTKDs*-KO HAP1 cells were measured by immunoblotting. γ-tubulin was used as a loading control for whole cell lysates. mtHSP70 and VDAC1 were used as mitochondrial loading controls. A representative result from two independent experiments is shown.

(mitochondrial myopathy, encephalopathy, lactic acidosis, and stroke)-like syndrome [31–34]. All other members of the FASTK family have been found to alter the expression levels of various mitochondrial mRNAs with some degree of specificity. While they regulate the abundance of specific mRNAs (Fig 1A) [16,18,19,24,26,27], others are also involved in the non-canonical mitochondrial RNA (mtRNA) processing events: mitochondrial FASTK (mtFASTK) and FASTKD4, for example, are specifically involved in the processing of pre-*ND6* and *ND5-CYTB* transcripts, respectively [16,18]. In contrast, FASTKD5 is required for the processing of all the non-canonical junctions with the exception of the 3′ UTR of *ND6* [19].

Despite their functional diversity, the FASTK family proteins share three putative RNA binding domains: FAST_1, FAST_2, and the RAP domain (Fig 1A) [22]. We and others have reported that the RAP domain seems structurally related to PD-(D/E)-XK nuclease superfamily [18,22,35], suggesting that this domain in the FASTK proteins may have endonuclease activity. Furthermore, RAP domain-containing proteins have been found to participate in nuclear and chloroplast RNA processing or splicing [36–39]. For instance, in *C. reinhardtii*, the chloroplastic RAP-domain-containing Raa3 protein participates in the trans-splicing of *psaA* mRNA that encodes a protein involved in photosynthesis [40]. Importantly, during trans-splicing of the *psaA* mRNA, Raa3 forms a complex with the FAST_1 domain-containing protein, Raa1 [41], supporting the hypothesis that these domains in the FASTK family members could be directly involved in mtRNA processing [18].

Here, we have investigated the role of different FASTK family proteins in mtRNA processing. We generated human cell lines with single or combined disruption of *FASTKD3*, *FASTKD4*, and *FASTKD5*. Transcriptomic analyses of these knockout cells demonstrated defective processing of several canonical and non-canonical junctions, accompanied by specific increases in particular antisense transcripts. Our results support the hypothesis that the FASTK protein family members are critical regulators of mitochondrial RNA processing.

## Results

### The FASTK family proteins specifically regulate the levels of sense and antisense transcripts and non-canonical mtRNA processing

To obtain greater insight into the role of FASTK family members in mitochondrial post-transcriptional regulation, we generated human cell lines in which either a single member of the family or a combination of members were knocked out. Knockout (KO) cells were generated in the HAP1 haploid cell line using CRISPR/Cas9 genome editing, and the efficiency of each KO was assessed by immunoblotting (Fig 1B), Sanger sequencing (S1 Fig), and RNA sequencing (RNA-seq) (S2 Fig). We generated single KO cell lines for *FASTKD3* (*FASTKD3*-sKO), *FASTKD4* (*FASTKD4*-sKO) and *FASTKD5* (*FASTKD5*-sKO), and two double KO cell lines (*FASTKD3&4*-dKO and *FASTKD4&5*-dKO) (Figs 1B and S1). It should be noted that subsequent analysis of the cell line in which *FASTKD3* and *FASTKD4* genes were deleted also carried a mutation in one of the *FASTKD5* alleles. However, this did not result in a significantly decreased expression of FASTKD5 protein (Fig 1B), and we therefore considered this cell line as a *FASTKD3&4*-dKO.

Importantly, we found that the HAP1 cell line expresses all members of the family except FASTKD2, which is absent both at the mRNA (S3B Fig) and protein levels (S3A Fig). The reason for this lack of FASTKD2 expression specifically in this cell line is currently unknown.

We then characterized these different KO cell lines by immunoblotting to investigate the expression of other FASTK family proteins and known mitochondrial RNA binding proteins (mtRBPs) involved in post-transcriptional regulation of mitochondrial gene expression (Fig 1B). FASTKD5 protein expression was decreased in *FASTKD3*-sKO cells and FASTKD3 protein level was downregulated in *FASTKD5*-sKO cells (Fig 1B), suggesting that the stability of these proteins may depend on their interaction, as has been shown for several other proteins, such as prohibitin 1 and 2 [42,43] and the mitochondrial pyruvate carrier subunits 1 and 2 [44], which are only stable as heterodimers. Furthermore, the absence of *FASTKD5* resulted in a substantial decrease in the MRPP1, ELAC2, and RPUSD4 proteins and an increase in the SUV3 protein as compared to controls (Fig 1B).

In the light of these results, we analyzed the different KO cell lines for mtRNA expression. We carried out northern blotting to measure the steady-state levels of mitochondrial

transcripts, including non-coding RNAs (ncRNAs), in control and the *FASTKDs*-KO cell lines (Figs 2 and S4). Of note, all the analyzed antisense or ncRNAs appeared as distinct bands with different lengths, indicating that they are processed in a specific manner before degradation. Furthermore, some ncRNAs were clearly processed into shorter ncRNAs, which we refer to as small non-coding RNAs (sncRNAs) (Figs 2A and S4).

In the *FASTKD3*-sKO cells, the level of *ND2*, *ATP8/6*, and *ND3* mRNAs derived from the HSP were significantly increased compared to controls (Fig 2A and 2B), consistent with our previous study [27]. However, LSP-derived transcripts were not particularly affected in these cells (Fig 2C).

Loss of *FASTKD4* caused the reduction of *ATP8/6*, *CO3*, *ND3*, *ND5*, and *CYTB* mRNAs as well as impaired *ND5-CYTB* RNA processing (Fig 2A, 2B and 2D), consistent with previous findings [18,26]. In contrast to these HSP-derived transcripts, several transcripts derived from LSP, *ND6*, *ncATP8/6*, and *ncCO1*, were accumulated in the absence of *FASTKD4* (Fig 2C). All sncRNAs except *sncCYTB* were decreased in the *FASTKD4*-sKO cells (Fig 2E), which is in contrast to the accumulation of their cognate longer ncRNAs. This finding suggests that FASTKD4 could be involved in the downstream processing of sncRNAs, in particular, *sncATP8/6* and *sncCO1*.

Loss of *FASTKD5* resulted in a significant accumulation of *ND1*, *ND2* and *CO2* mRNAs and a decrease in *CO1*, *ATP8/6*, *CO3*, and *ND5* mRNAs (Fig 2A and 2B). Furthermore, processing of pre-*CO1*, *ATP8/6-CO3*, and *ND5-CYTB* non-canonical precursor transcripts, was impaired, as previously reported [19], and this defect was particularly striking for pre-*CO1* and *ATP8/6-CO3* RNA processing, resulting in very low levels of mature mRNAs (Fig 2A and 2D). By comparison to *FASTKD4*-sKO cells, the processing defect of *ND5-CYTB* in the *FASTKD5*-sKO cells was subtle. This indicates that *ND5-CYTB* RNA processing is mainly regulated by FASTKD4. On the other hand, among the transcripts derived from LSP, only *ncATP8/6* RNA dramatically accumulated when *FASTKD5* was absent (Fig 2A and 2C). It is important to note that the *sncATP8/6* RNA level was reduced in the *FASTKD5*-sKO cells, as it was in the *FASTKD4*-sKO cells. This finding suggests that FASTKD4 and FASTKD5 cooperate in the processing of *sncATP8/6* RNA. It is noteworthy that the HSP-derived transcripts, *ND2*, *CO2*, and *ND3* mRNAs, accumulated in either *FASTKD3*-sKO or *FASTKD5*-sKO cells (Fig 2B). However, the molecular phenotypes of the two cell lines showed some distinct changes. If, as proposed above, the stability of these two proteins was interdependent, one would expect similar phenotypes for both KO cells. The fact that there is residual FASTKD5 in *FASTKD3*-sKO and some FASTKD3 in *FASTKD5*-sKO cells may explain the observed differences between the two cell lines.

Next, we investigated mtRNA expression in the double KO cells. *FASTKD3&4*-dKO resulted in a decrease in *ATP8/6*, *CO3*, *ND3*, *ND5*, and *CYTB* mRNAs with impaired *ND5-CYTB* RNA processing (Fig 2A, 2B and 2D), which is similar to the phenotype of *FASTDK4*-sKO cells. Notably, the *ND3* mRNA level, which is antagonistically regulated by FASTKD3 and FASTKD4, was decreased in the *FASTKD3&4*-dKO cells, as it was in the *FASTKD4*-sKO cells. Interestingly, LSP-derived transcripts were not significantly affected in the *FASTKD3&4*-dKO cells, in contrast to the single *FASTKD4*-sKO cells (Fig 2C). However, *sncCO2* and *sncCO1* RNA levels were most decreased in the dKO cells, similar to in the *FASTKD4*-sKO cells (Fig 2E), further supporting the essential role of FASTKD4 in the generation of *sncCO1* RNA.

Double knockout of *FASTKD4* and *FASTKD5* caused a decrease in *CO1*, *ATP8/6*, *CO3*, *ND3*, *ND5*, and *CYTB* mRNAs with defective processing of pre-*CO1*, *ATP8/6-CO3*, and *ND5-CYTB* precursor RNAs (Fig 2A, 2B and 2D). This phenotype is therefore the result of the combined effects of the sKOs, which suggests that these two proteins may play overlapping or cooperative functions. This is particularly striking for the *ND5-CYTB* RNA processing, which was completely absent in the dKO cell line (Fig 2A). However, the *ND1* and *ND2* mRNA levels,

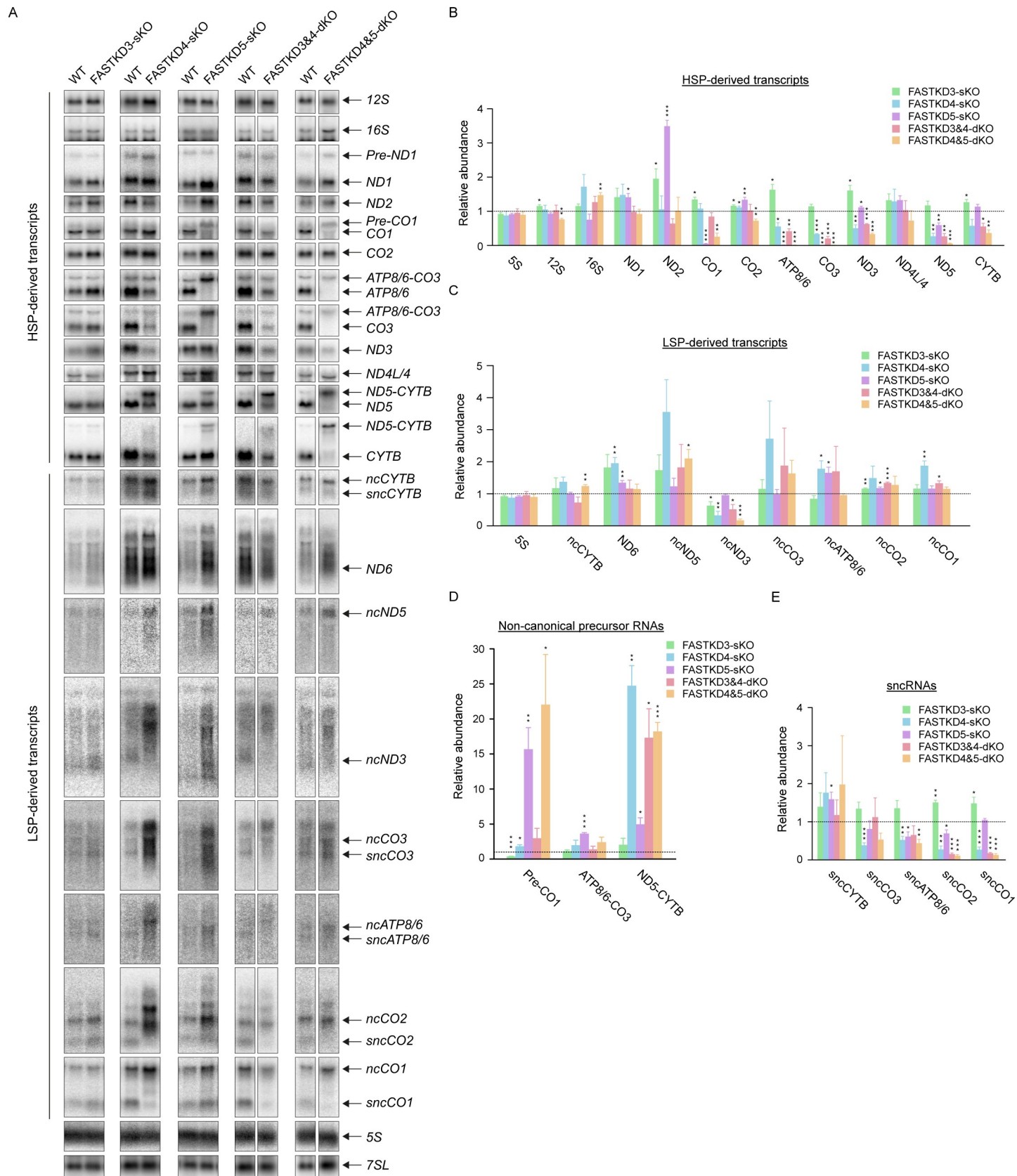

**Fig 2. Loss of FASTKD proteins alters the abundance of coding and non-coding RNAs and perturbs non-canonical RNA processing.** (**A**) The steady-state levels of rRNAs, mRNAs and ncRNAs in WT (control) and *FASTKDs*-KO HAP1 cells were measured by northern blotting. *5S* and *7SL* were used as loading controls. The equal amount of total cellular RNA was loaded on several separate gel. Each comparison (WT vs KO) shown is from the same gel. A lane of WT is exceptionally used for both the pair of *FASTKD4*-sKO and the pair of *FASTKD3&4*-dKO. A representative result from three independent experiments is shown. (**B-E**) Relative abundance of mature rRNAs, mRNAs, ncRNAs derived from either the heavy strand (B) or the light strand (C), non-canonical precursor RNAs (D), and sncRNAs (E). The data from each of the *FASTKDs*-KO HAP1 cell lines were quantified by comparison to WT HAP1 cells (control) (n = 3, except for Fig 2D with n = 2 or 3). RNA loading was normalized to the content of *7SL* RNA. The dotted line shows the relative abundance of control (= 1). Results are mean±SEM, *P<0.05, **P 0.001 to 0.01, ***P <0.001 (two-tailed unpaired t-test).

which were increased in the *FASTKD5*-sKO cells, were comparable to the controls in the *FASTKD4&5*-dKO cells as well as in the *FASTKD4*-sKO cells (Fig 2B). *ncND5* was the only LSP-derived transcript that accumulated significantly in the *FASTKD4&5*-dKO cells (Fig 2C). The *FASTK4&5*-dKO additionally resulted in a dramatic decrease in *sncATP8/6*, *sncCO2*, and *sncCO1* RNAs (Fig 2E). In contrast to our findings in the *FASTKD4*-sKO cells, the *FASTKD3&4*-dKO and the *FASTKD4&5*-dKO cell lines only showed mild accumulation of ncRNAs, suggesting that the ncRNA levels could be compensated by other mtRBPs when multiple FASTKD proteins are lost.

We observed that the overall mitochondrial transcription level, as assessed by the level of *7S* RNA, which is the most LSP-proximal transcript, appeared normal in all the KO cell lines (S3C Fig). Moreover, mtDNA abundance was also not altered in any of the KO cell lines (S3D Fig), consistent with the unchanged level of LSP transcription that forms a DNA replication primer (S3C Fig) [45]. In addition, we carried out SSBP1 and BrU staining to assess mitochondrial DNA replication and transcription respectively, and in both cases, we found similar levels of immunostaining in all the KO and control cells (S3E Fig). These findings are therefore entirely consistent with the hypothesis that the FASTK family proteins are involved in post-transcriptional regulation, and they fully support our conclusions from the northern blotting analyses (Fig 2) that the regulation of mtRNA abundance is a consequence of post-transcriptional events.

It is notable that the accumulation of ncRNAs in the different *FASTKD*-KO cells was often inversely correlated with the change in their complementary mRNAs (Fig 2B and 2C). For instance, the antisense transcript of *ATP8/6* accumulated following the loss of either *FASTKD4* or *FASTKD5*, whereas its complementary *ATP8/6* mRNA was decreased in these sKO cells. This suggests the possibility that the FASTKD proteins could regulate the levels of specific mRNAs through the processing and degradation of their antisense ncRNAs. This process would probably involve recruitment of the mtRNA degradosome, and indeed, the pattern of ncRNAs in the *FASTKD4*-sKO cells is reminiscent of the phenotype of cells lacking PNPase, the catalytic enzyme of the mtRNA degradosome [46]. Furthermore, PNPase was recently found to prevent the accumulation of mitochondrial double-stranded RNA (dsRNA) [47], which may potentially inhibit mtRNA processing due to duplex formation. Mitochondrial dsRNA can be quantified using the anti-dsRNA J2 antibody [47,48]. However, we found that in the *FASTKD4*-sKO cells, the J2 immunostaining, as well as the quantification of mitochondrial dsRNA by dot blotting, did not change compared to controls (S3E and S3F Fig). Nonetheless, since the J2 antibody only recognizes dsRNA that is longer than 40 nucleotides, we cannot rule out that shorter dsRNA molecules might accumulate in *FASTKD4*-sKO cells.

## Mitochondrial transcriptomics demonstrates the specific roles of the FASTKD proteins in stability and processing of both sense and antisense transcripts

In a complementary study, we carried out RNA-seq using the procedures we have reported previously [12,13,29] to analyze the effects of the different *FASTKD* gene KOs on the

mitochondrial transcriptome. We first investigated the abundance of rRNAs and mRNAs in each of the *FASTKDs*-KO cell lines (Fig 3A). In most cases, the RNA-seq data confirmed the results of northern blotting, although some exceptions were noted: the RNA-seq data showed that loss of *FASTKD4* additionally caused a reduction in *ND1*, *ND2*, *ND4L/4* and *ND6* mRNAs, and the *FASTKD3&4*-dKO caused a reduction in *ND1*, *ND2*, and *ND6* mRNAs, while the *FASTKD4&5*-dKO resulted in an increase in the *16S* rRNA, which was not observed in either sKO. These findings, together with the northern blot data (Fig 2), indicate that the loss of *FASTKD4* and *FASTKD5* have the most significant effects on the stability of specific mRNAs.

We next analyzed the entire mitochondrial transcriptome for processing defects as a result of different *FASTKD* gene knockouts. The RNA-seq library construction approach enabled us to exclude mature tRNAs and retain unprocessed transcripts containing tRNAs that intersperse adjacent coding regions in the mitochondrial transcriptome. In the *FASTKD3*-sKO cells, there were no processing defects for any heavy strand-encoded RNAs indicating that this protein does not play a major role in mitochondrial RNA cleavage (Fig 3B). However, its loss resulted in specific increases in $tRNA^E$- and $tRNA^Q$-containing LSP-derived precursor RNAs. In the *FASTKD4*-sKO cells, processing defects were observed at the junction between *ND2* and *CO1* and at the *ND5-CYTB* junction. In addition, there was a processing defect at the region between *ND1* and *ND2* (Fig 3B), consistent with the reduction of these two mRNAs (Fig 3A). In the *FASTKD5*-sKO cells, the processing defect at the *ATP8/6-CO3* junction was most pronounced (Fig 3B), consistent with the northern blotting data (Fig 2B and 2D). However, we also observed that in the absence of *FASTKD5* the processing of the *ND5-CYTB* junction and the 5′ UTR of *CO1* were also impaired (Fig 3B).

An investigation into the effects of the *FASTKD3&4*-dKO on RNA processing revealed processing defects at the junction between *ND2* and *CO1* and at the *ND5-CYTB* junction, similar to the *FASTKD4*-sKO (Fig 3B), indicating that the primary processing role of FASTKD4 may be at these non-canonical junctions. Interestingly, the processing defect found at the junction between *ND1* and *ND2* in the *FASTKD4*-sKO cells, was absent in the *FASTKD3&4*-dKO, suggesting that the loss of *FASTKD3* might compensate for the absence of *FASTKD4* at this cleavage site. These findings thus provide evidence for partially overlapping roles of the FASTKD proteins in mtRNA processing. The greatest enrichment of unprocessed RNA precursors was observed in the *FASTKD4&5*-dKO cells; the most prominent of these defects was at the junction between *ND5* and *CYTB*, while significant changes were also seen in the *ND1-ND2* junction, the 5′ UTR of *CO1*, the *ATP8/6-CO3* junction, and the junction between *ND4L/4* and *ND5* (Fig 3B). These findings indicate that both FASTKD4 and FASTKD5 play a role in mtRNA processing at the *ND1-ND2*, *ND4L/4-ND5* and *ND5-CYTB* junctions consistent with the data from northern blotting (Fig 2). On the other hand, the dKO cell lines revealed that FASTKD5 is exclusively responsible for the specific cleavage of the non-canonical junctions at the *ATP8/6-CO3* junction and at the 5′ UTR of *CO1*. Rapid amplification of cDNA ends (RACE) and sequencing confirmed that cleavage at the 5′ UTR of *CO1* requires FASTKD5, since the precursor *CO1* transcript was captured in the *FASTKD5*-sKO and *FASTKD4&5*-dKO cells but not in the WT and *FASTKD4*-sKO cells (S5A and S5B Fig). The control 5′ end of *CO2* was unaffected in all tested cell lines (S5C Fig). This finding was further validated by northern blotting against the antisense $tRNA^Y$, that precedes the *CO1* transcript, which showed an increase of the precursor *CO1* transcript in cells lacking *FASTKD5* compared to the WT and *FASTKD4*-sKO cells (S5D Fig). Similarly, the precursor *CO1* transcript was present in cells lacking *FASTKD5*, while mature *CO1* mRNA was only detected in the WT and *FASTKD4*-sKO cells (S5D Fig). Furthermore, the uncleaved precursors that were observed at the *ATP8/6-CO3* junction in the *FASTKD5*-sKO cells were reduced in the *FASTKD4&5*-dKO cells.

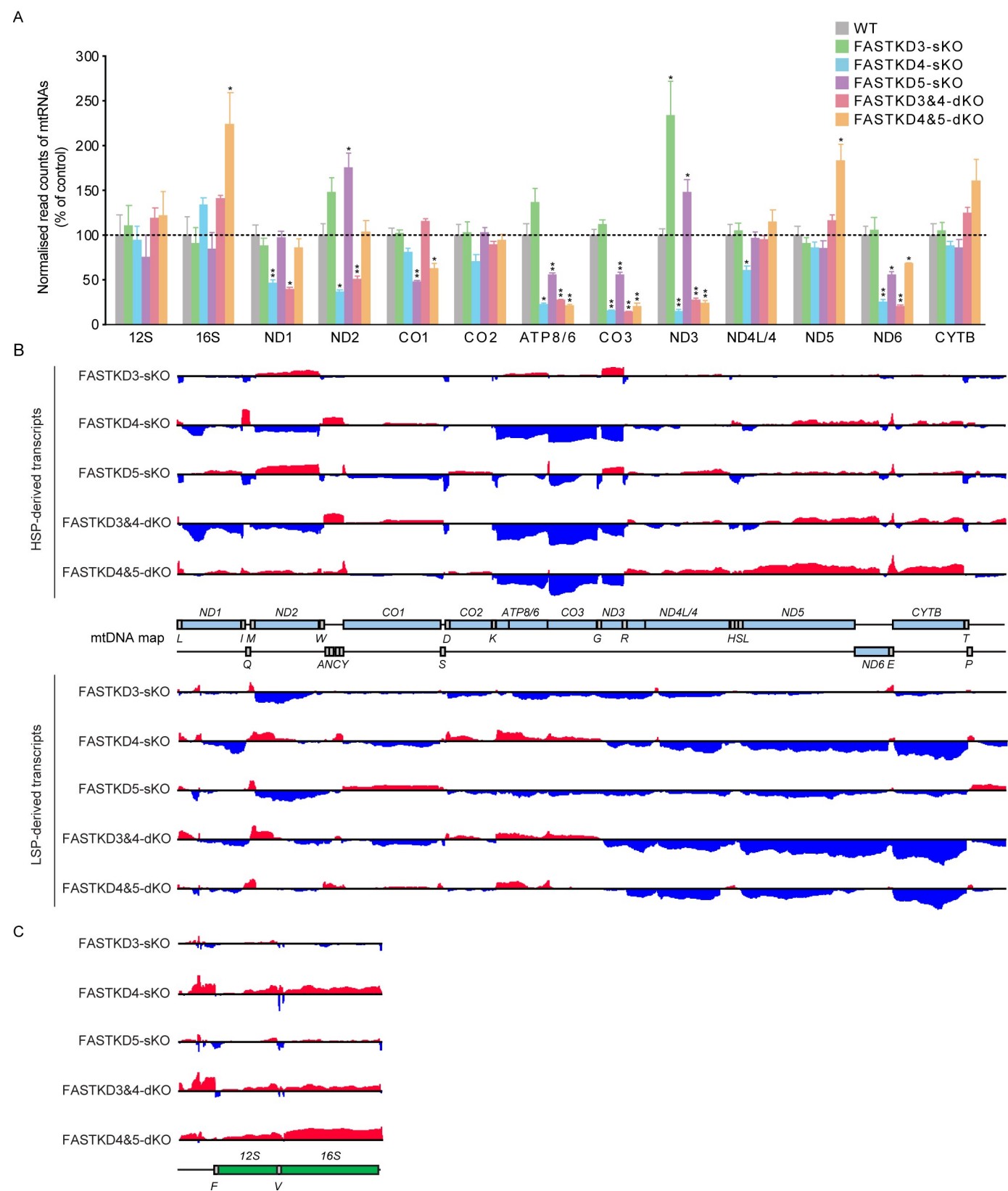

**Fig 3. Mitochondrial transcriptome analysis of the *FASTKDs*-KO cells by RNA-seq.** (**A**) Relative abundance of read counts normalized to total reads for mitochondrial mRNAs were analyzed. Results are mean±SD *P<0.05, **P<0.01 (Wald test). (**B-C**) A complete map of mtRNA abundance and mean (log₂ fold

change [KO$_{mean}$/Ctrl$_{mean}$]) determined by RNA-seq coverage from control, *FASTKD3*-sKO, *FASTKD4*-sKO, *FASTKD5*-sKO, *FASTKD3&4*-dKO, and *FASTKD4&5*-dKO cells (n = 3). Increases in coding regions (B) and rRNA regions (C) are shown in red, and decreases in coding regions (B) and rRNA regions (C) are shown in blue. The range of the log$_2$ fold changes are from -3 to 2.8.

It is interesting to note that processing of the normally tRNA-punctuated junctions between the rRNAs was not affected by the loss of any of the *FASTKD3-5* proteins (Fig 3C), suggesting that these proteins likely participate in the downstream processing of non-canonical junctions, and possibly act downstream of MRPP3 and ELAC2. Their loss may nevertheless stimulate a general response to defective processing because more efficient processing is observed at certain canonical MRPP3/ELAC2 processing sites, such as *tRNA$^T$*, in their absence (Fig 3B).

These data also allowed us to investigate how the loss of each of the FASTKD3-5 proteins affected the non-coding regions of the mitochondrial transcriptome (Fig 3B). Loss of *FASTKD4* resulted in the accumulation of transcripts or junction-containing RNAs that are antisense to the *ND2*, *CO2*, *ATP8/6* and *CO3* mRNAs, whereas loss of *FASTKD5* caused the accumulation of the *CO1* antisense RNA (Fig 3B). The *FASTKD3&4*-dKO cells resulted in an accumulation of the same ncRNAs found in the *FASTKD4*-sKO cells, suggesting that FASTKD4 plays the predominant role in the removal of these antisense transcripts. In the case of the antisense *ATP8/6*, this is the only transcript that accumulates in the *FASTKD4&5*-dKO cells but not in the *FASTKD5*-sKO cells. The increase of specific ncRNAs is in contrast to the reduction of their complementary mRNAs and may suggest that the FASTKD proteins have a role in the clearance of the antisense transcripts during RNA processing and that in the absence of FASTKD proteins, the antisense transcripts accumulate. Impaired clearance of antisense RNAs can cause processing defects and destabilization of their complementary mRNAs since the accumulated antisense transcripts may allow duplex formation, as we have shown previously [49]. This duplex potentially precludes the processing of mRNAs and reduces their translation.

To investigate further the consequences of impaired processing of mtRNAs in the absence of the FASTKD proteins, we analyzed the effects on polyadenylation by mapping RNAseq reads to polyadenylated mRNA reference sequences identify changes in polyadenylation (Figs 4 and S6). We show that the polyadenylation is not affected in mRNAs, such as *CO2*, whose levels were not changed by the loss of specific FASTKD proteins, either in single or double KOs (Fig 4A). The polyadenylation status was also not affected in specific mRNAs, such as *ATP8/6* (Fig 4B), even though its levels were changed in the KO cell lines compared to control cells, indicating that the loss of specific FASTKD proteins affects more the abundance rather than the polyadenylation status of these mRNAs. The polyadenylation of the *ND5* mRNA without its 3' UTR was reduced in the *FASTKD4* and *FASTKD5* KO cells (Fig 4C), but not the polyadenylation status of the *ND5* mRNA with its 3' UTR and the *CYTB* mRNA (S6 Fig).

## FASTKD4 specifically binds to *tRNA$^E$*, which is complementary to the *ND5-CYTB RNA* processing site

The transcriptome analysis of the *FASTKDs*-KO cell lines indicated that the FASTKD proteins are involved principally in the processing of non-canonical RNA junctions. Furthermore, these proteins, especially FASTKD4, may play a role in the elimination of ncRNAs. To understand how FASTKD4 could achieve its function, we performed a HITS-CLIP analysis to identify which RNA species are bound by FASTKD4 (Figs 5, S7A, and S7B). Using endogenous FASTKD4 as bait we found that FASTKD4 preferentially associates with mitochondrial *tRNA$^E$*, which is encoded on the strand, opposite the *ND5-CYTB* junction (Figs 5A, S7A, and S7B). This preferential binding was confirmed by RT-qPCR (S7C Fig). Using UV cross-linked

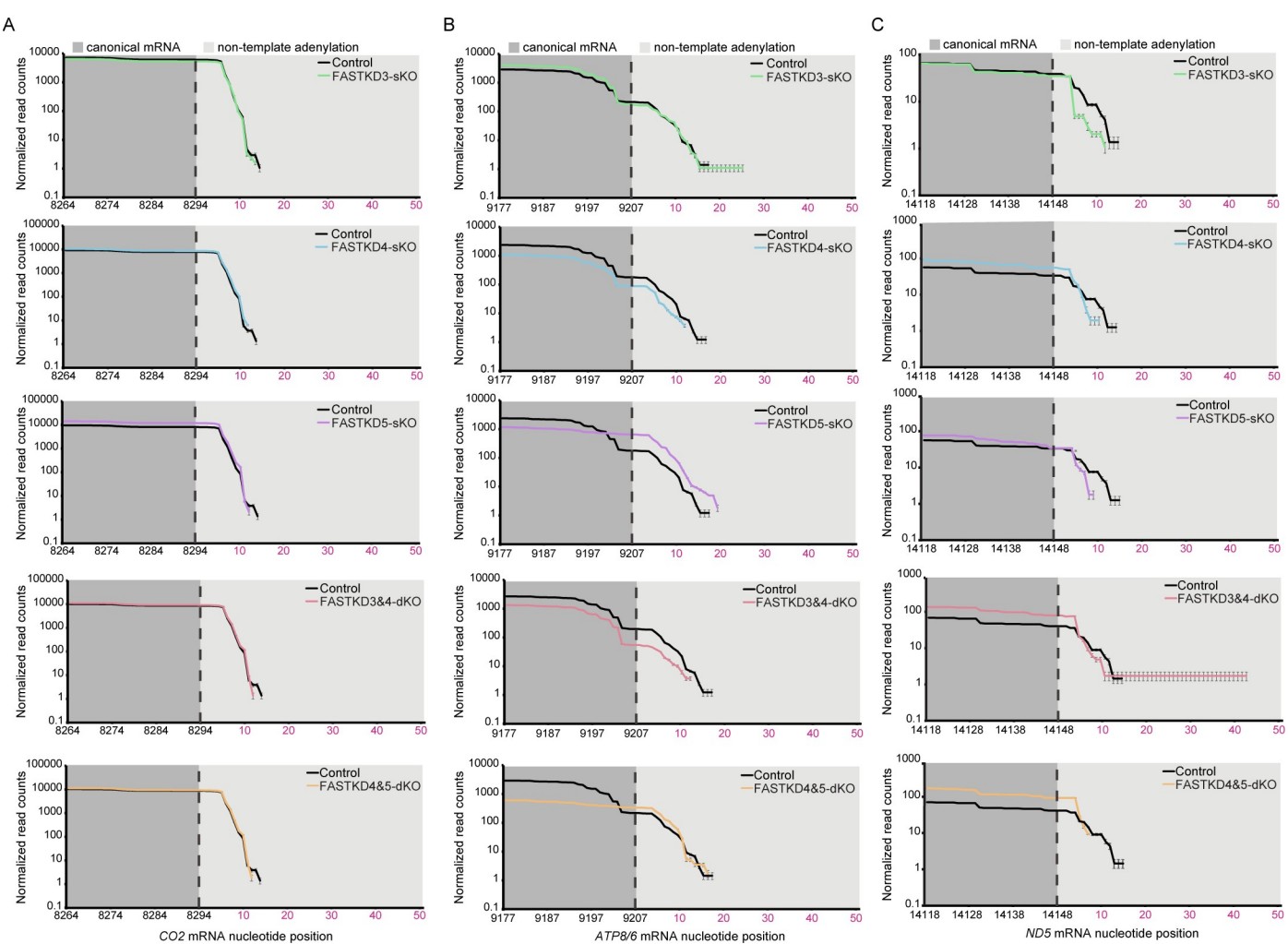

**Fig 4. Polyadenylation status of mitochondrial mRNAs in *FASTKDs-KO* cells.** Mapping of RNA-seq reads to polyadenylated mitochondrial mRNA reference sequences reveals no changes in *CO2* (**A**) or *ATP8/6* (**B**) mRNAs but a reduction in polyadenylation of *ND5* mRNA was identified in the *FASTKD4* and the *FASTKD5* KO cell lines (**C**), compared to control cells. The numbering on the x-axis shows the mtDNA position of the transcript sequence and the canonical 3' region of each mRNA is shown in dark grey. The predicted canonical cleavage site is marked with the dotted line, followed by numbering of the non-templated poly (A) residues shown in the light grey shading. The y axis shows the normalized read counts at each position.

WT and *FASTKD4*-sKO HAP1 cells, we immunoprecipitated FASTKD4 from isolated mitochondria, purified the RNA from the immunoprecipitate, and performed RT-qPCR for various transcripts. We confirmed that mitochondrial *tRNA^E^* was enriched in the FASTKD4 immunoprecipitate (S7C Fig). However, we noticed that in addition to the *tRNA^E^*, other transcripts were also enriched which did not score positive in the HITS-CLIP analysis. This suggests that FASTKD4 associates more generally with unprocessed polycistronic transcripts in addition to the *tRNA^E^*, or possibly also interact with other transcripts. Curiously, we observed that the *tRNA^E^* level was not altered in *FASTKD4*-sKO cells even though the *ND5-CYTB* precursor on the heavy strand was shown to accumulate (S3G Fig).

The identification of preferential binding of FASTKD4 to the *tRNA^E^* is an intriguing result, given that this RNA is within the complementary sequence on the opposite strand of the *ND5-CYTB* junction, the processing of which is known to be impaired in the loss of *FASTKD4*. Around 40% of the HITS-CLIP reads covering *tRNA^E^* included the 3' end of its coding

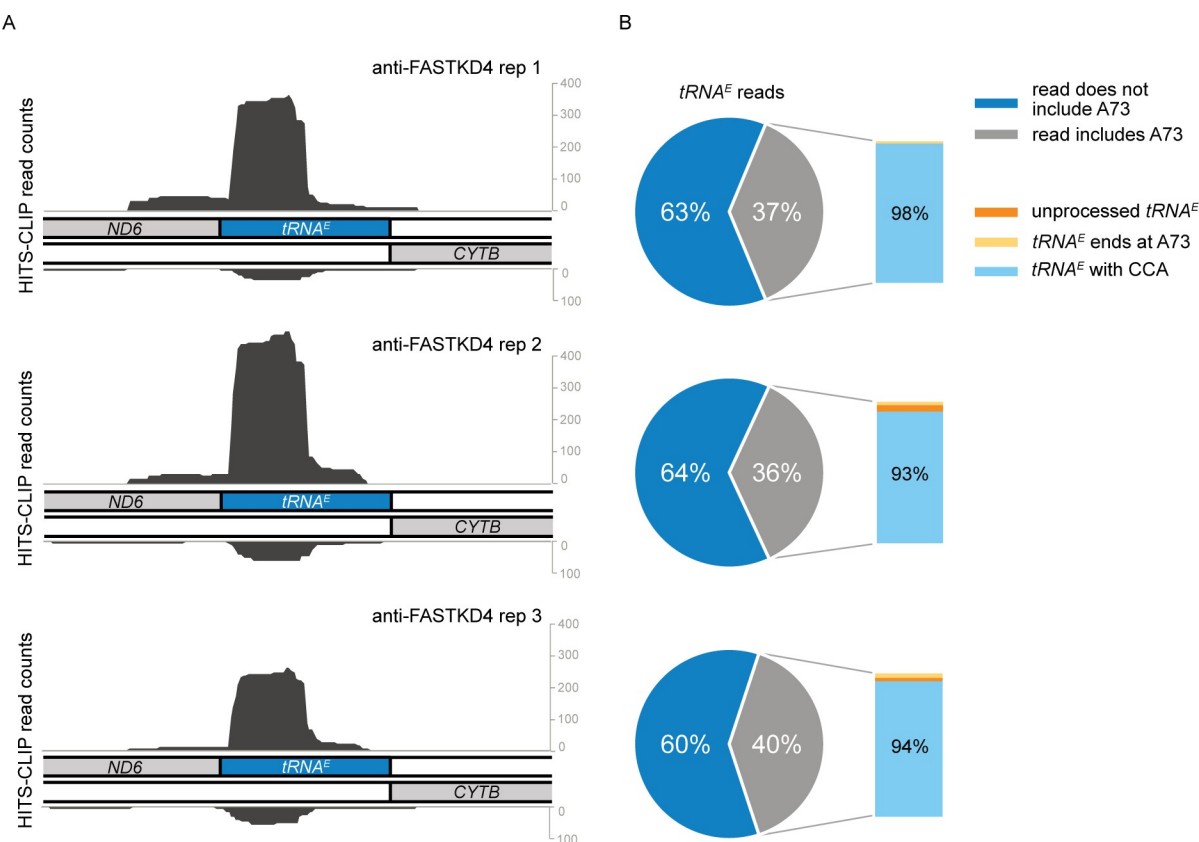

**Fig 5. FASTKD4 specifically binds to *tRNA^E* complementary to the *ND5-CYTB* RNA processing site.** (**A-B**) Strand specific HITS-CLIP read counts corresponding to the 14600–14800 region of the mtDNA for anti-FASTKD4 antibody (A). The pie-chart in (B) shows the fraction of *tRNA^E* reads that includes the last encoded position (A73) of *tRNA^E* (grey) and reads that are truncated on the 3′ end of *tRNA^E* (dark blue). The inset shows all reads that include A73 of *tRNA^E* with the fraction of unprocessed RNAs containing an extended sequence (orange), the fraction of reads that end exactly at A73 (yellow), and the fraction of those that have a CCA addition (light blue).

sequence, namely A73, and the majority of these A73-containing reads additionally included a 3′ extension with the CCA tail (Fig 5B). This indicates that FASTKD4 can interact with the post-transcriptionally modified, CCA-containing mature *tRNA^E*. However, the time course of this binding is still uncertain, as we do not know whether FASTKD4 binds *tRNA^E* before or after its processing and maturation. Nevertheless, the binding of FASTKD4 to this specific region provides a mechanistic explanation for the role of FASTKD4 in the processing of the *ND5-CYTB* junction, as discussed later.

## The FASTK family proteins are essential for expression of the mitochondrially encoded respiratory complex proteins

To investigate further the phenotypes of the *FASTKDs*-KO cell lines, we measured *de novo* mitochondrial translation in the presence of $^{35}$S-methionine and $^{35}$S-cysteine (Fig 6A). Loss of *FASTKD3* caused a specific increase in ND3 protein synthesis compared to controls, which is consistent with an increase of *ND3* mRNA (Figs 2B and 3A). *FASTKD4*-sKO cells showed a decrease in expression of the ND3 and COX3 proteins also in line with their mRNA levels. However, levels of newly synthesized ND5 and COX1 proteins were increased following the loss of *FASTKD4* despite the low level of mature *ND5* mRNA (Figs 2B, 2D, and 3B), and this was also found in the *FASTKD3&4*-dKO cells (Fig 6A). The reason for the poor correlation

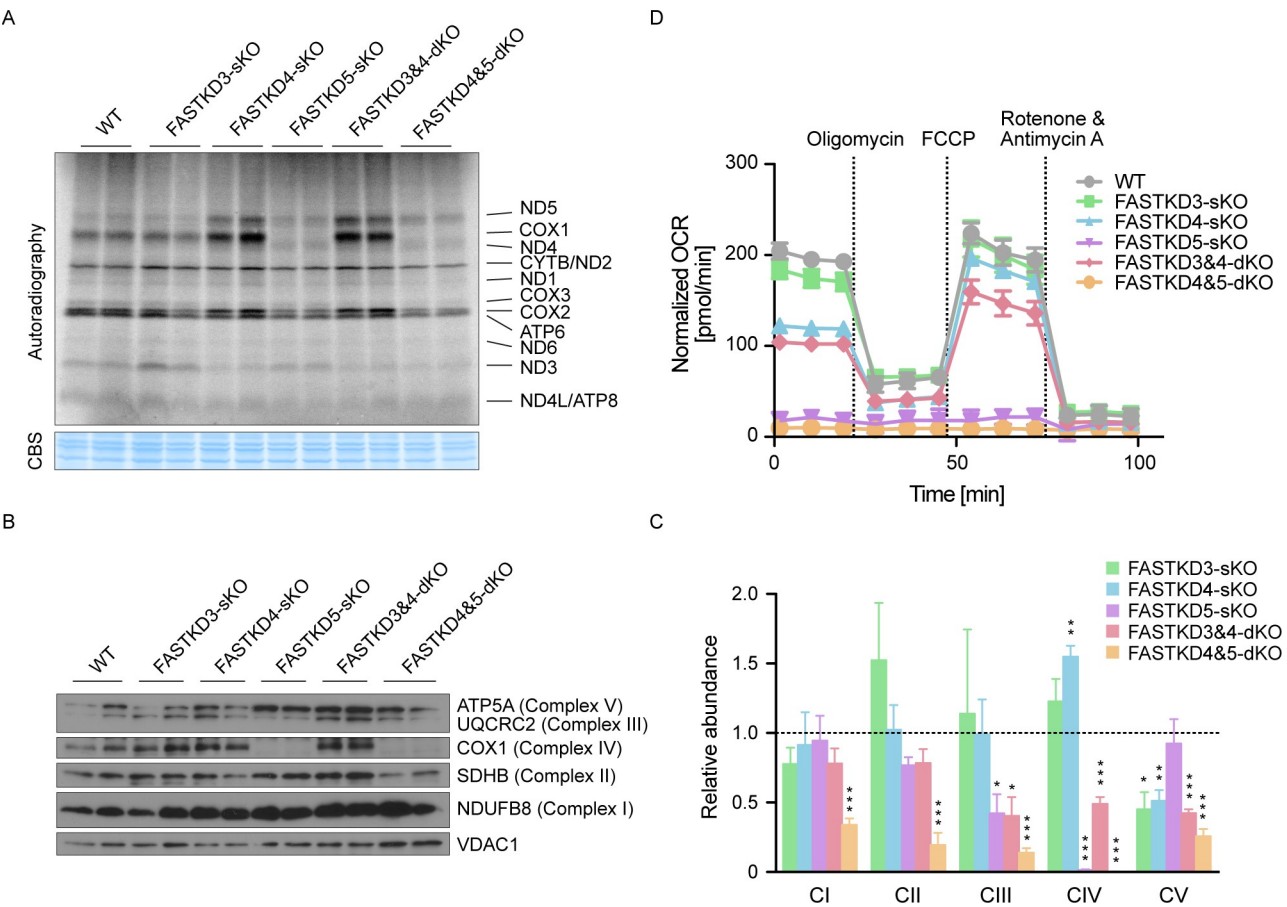

**Fig 6. Effect of FASTK family protein depletion on mitochondrial translation, respiratory chain assembly and oxygen consumption. (A)** Levels of *de novo* mitochondrial protein synthesis were measured in WT (control) and *FASTKDs*-KO HAP1 cells by pulse incorporation of [35]S-labeled methionine and cysteine. Equal amounts of whole cell lysates were separated by SDS-PAGE and visualized by autoradiography. **(B)** Protein expression levels of specific subunits of each of the MRC complexes in WT (control) and *FASTKDs*-KO cells were evaluated by immunoblotting. Total OXPHOS Rodent WB Antibody Cocktail antibody was used to assess the efficiency of the MRC complex assembly. VDAC1 was included as a mitochondrial loading control. **(C)** Relative abundance of each assembled MRC complex was determined by quantification of the results of immunoblotting (n = 3). Loading amounts were normalized relative to VDAC1. The dotted line shows the relative abundance of control (= 1). Results are mean±SEM, *P<0.05, **P 0.001 to 0.01, ***P <0.001 (two-tailed unpaired t-test). **(D)** Oxygen consumption rate (OCR) in WT (control) and *FASTKDs*-KO cells was measured using the Seahorse analyzer. OCR was normalized to the number of cells in each well. A representative result from three independent biological experiments is shown.

between mRNA level and protein translation in this case is currently unclear. On the other hand, we do not understand why COX1 translation is increased in the absence of *FASTKD4*, given that the *CO1* mRNA level is not affected. This suggests that in addition to playing some role in RNA processing, FASTKD4 may also participate in the control of translation of specific mRNAs as it associates with factors that modulate RNA metabolism and translation such as LRPPRC and IARS2 (S8 Fig). Finally, the absence of *FASTKD4* was found to normalize the level of *ND3* mRNA (Figs 2B and 3A) and protein (Fig 6A), which was upregulated in *FASTKD3*-sKO cells. This is consistent with our earlier conclusion that, at least in this specific case, FASTKD4 can compensate for the loss of *FASTKD3*.

Loss of *FASTKD5* caused a reduction in protein synthesis of COX1, COX3 and ATP6 (Fig 6A) consistent with their decreased mRNA levels, as well as reduced COX2 translation, which was not correlated with a change in mRNA abundance (Figs 2B and 3A). Remarkably, COX1 protein synthesis was strongly perturbed in the cells lacking *FASTKD5*, consistent with the

severe defect in pre-*CO1* RNA processing (Figs 2B, 2D, and 3B). ND5 protein synthesis was not affected in the *FASTKD5*-sKO, which probably reflects the minor involvement of FASTKD5 in the processing of the *ND5-CYTB* junction (Figs 2A, 2D, and 3B). It is striking that, while the correctly processed *ND5* mRNA level is almost undetectable in the *FASTKD4&5*-dKO cells (Fig 2), the level of ND5 protein was normal (Fig 6A). These results support the hypothesis that, at least for ND5, the protein may be translated from the *ND5-CYTB* precursor RNA.

Consistent with these results, using an antibody cocktail that recognizes subunits from all mitochondrial respiratory complexes (MRC) to assess their abundance, we found that Complex IV was most affected by the loss of *FASTKD5* (Fig 6B and 6C). This result was validated by oxygen consumption measurements of the different KO cells (Fig 6D). We found that both *FASTKD4*-sKO and *FASTKD3&4*-dKO cells had only minor defects in OXPHOS. In contrast, oxygen consumption was dramatically reduced in both the *FASTKD5*-sKO and the *FASTKD4&5*-dKO cells. These findings indicate that FASTKD5 is indispensable for OXPHOS because it is essential for COX1 protein synthesis and the assembly of Complex IV.

## Discussion

Post-transcriptional control of human mitochondrial gene expression is regulated in time and space, with many events occurring within the mitochondrial RNA granules [20,50]. Dysfunction of these post-transcriptional events can lead to numerous, often severe, mitochondrial pathologies [34,51]. Here, we have characterized the FASTK family proteins that are emerging as key post-transcriptional regulators of mitochondrial gene expression, and we have focused on their roles in mtRNA processing. Their function in non-canonical mtRNA processing is evolutionarily conserved. *C. elegans* is an ancestral model organism that has only one FASTK orthologue, FASK-1 (FASTK-related protein 1), to which mammalian FASTKD4 is the most closely related [18]. Deletion of the *fask-1* gene resulted in a specific accumulation of *ND5*-containing precursor transcripts (*16S-ND3-ND5* and *ND3-ND5*), which lack tRNA junctions, and a decrease in mature *ND5* mRNA (S9 Fig).

Mitochondrial transcriptome analyses of HAP1 cell lines carrying single or double knockouts of *FASTKD3-5*, revealed that FASTKD4 and FASTKD5 control the maturation and abundance of a broad range of mitochondrial mRNAs, and furthermore, they also affect the fate of many of the ncRNAs in the mitochondrial transcriptome. The loss of *FASTKD4* resulted in impaired *ND5-CYTB* RNA processing, and the absence of *FASTKD5* caused defects in *ND5-CYTB*, pre-*CO1*, and *ATP8/6-CO3* RNA processing, as previously reported [18,19]. None of these effects were seen in cells depleted of *FASTKD3*. The defective processing of the *ND5-CYTB* junction was observed in the single KOs of *FASTKD4* and *FASTKD5*, and the effect persisted in the double mutant *FASTKD4&5*-dKO, suggesting that FASTKD4 and FASTKD5 may cooperate to ensure optimal processing of this junction. However, the interactions between members of the FASTK family can be more complex since we revealed antagonistic activities in the processing, or stabilization of some mRNAs. For instance, the *ND3* mRNA level was significantly increased in the *FASTKD3*-sKO and *FASTKD5*-sKO cells but decreased in the *FASTKD4*-sKO cells compared to controls. Interestingly, in both dKO cell lines (*FASTKD3&4*-dKO and *FASTKD4&5*-dKO), the *ND3* mRNA level was reduced, indicating that FASTKD4 may play the predominant role in regulating the expression of the *ND3* mRNA. It is interesting to note that the processing of $tRNA^G$ and $tRNA^R$, which are tRNAs that flank *ND3*, was boosted in the *FASTKD3*-sKO and the *FASTKD5*-sKO, whereas this was not observed in the *FASTKD4*-sKO and the dKO cells, suggesting that FASTKD4 increases the stability of *ND3* mRNA after it is cleaved from its precursor transcript.

The complex interactions between FASTKD members are also illustrated by the interdependence of FASTKD3 and FASTKD5 in terms of their mutual stability. Loss of one member of the family leads to a lower expression of the other, suggesting a close physical interaction between them, as is found for other proteins that show a similar interdependence [42–44]. However, we did not detect an interaction between FASTKD3 and FASTKD5 in the BioID analysis (S8 Fig). It is important to note that this inconsistency may be cell-type specific effects since 143B cells, which we used in the BioID analysis, express FASTKD2 protein unlike HAP1 cells (S3A Fig), and the expression of FASTKD2 may dynamically reprogram protein interaction profiles of the FASTK protein family in these cells. Further experiments are required to test this hypothesis. On the other hand, the BioID analysis revealed a physical interaction between FASTKD4 and FASTKD5. These data taken together, provide evidence that FASTK family members may interact to form large protein complexes, capable of exerting a variety of regulatory functions, depending on the specific combination of the FASTKD, and possibly other protein components. Among the proteins affected by the loss of *FASTKD* family members, which may also participate in the formation of larger complexes, are the post-transcriptional tRNA processing and modifying enzymes, MRPP1 (TRMT10C), ELAC2 and RPUSD4. In addition, the BioID experiments using FASTKD4 and FASTKD5, identified several other interacting proteins such as LRPPRC and GRSF1, which could constitute further components of the protein machinery that is functionally linked to RNA processing (S8 Fig) [13]. This is supported by a recent study of mitochondrial protein interaction networks based on the BioID approach, which revealed that the FASTK family members and a subset of the mtRBPs, including those identified in this study, co-localize within the mitochondrial matrix [52].

*De novo* translation analyses revealed that the loss of the FASTKD proteins result in significant defects in mitochondrial translation and in mitochondrial respiration. Importantly, the extent of the defect in oxygen consumption was proportional to the effect on non-canonical RNA processing. These experiments on translation levels also revealed that the level of mRNA does not always reflect their translation efficiency. For instance, ND5 protein synthesis was normal or even higher in the *FASTKD4*-sKO and *FASTKD4&5*-dKO cells, despite the very low expression level of its mRNA. This suggests either a highly efficient translation of the residual *ND5* mRNA or the possibility that ND5 can be translated from the increased levels of the *ND5-CYTB* precursor transcripts. In addition, ND4 protein synthesis was elevated in cells lacking *FASTKD5*, regardless of its mRNA level. These findings raise the hypothesis that FASTK family proteins can modulate mitochondrial translation.

The mitochondrial transcriptome analyses revealed that the FASTKD proteins are also involved in the cleavage of certain tRNA junctions such as the cluster of tRNAs between *ND1* and *ND2* and $tRNA^E$ between *ncCYTB* and *ND6*. This implies that the FASTKD proteins may recruit the tRNA processing enzymes at some specific sites. However, the processing of some other tRNA junctions, for example $tRNA^D$ between *CO1* and *CO2*, were facilitated following the loss of the FASTKD proteins. These data show that the FASTKD proteins specifically facilitate or inhibit tRNA processing, which largely affects the metabolism of mRNAs.

The transcriptome analyses also indicated that FASTKD5 seems to be involved in the cleavage of the non-canonical junction between antisense *ATP8/6* and *CO3*. In this case, a mechanism other than the recruitment of known RNA processing enzymes must be responsible, as knockout of MRPP3 and ELAC2 does not affect this site [11–13]. For this event, FASTKD5 could exhibit its own intrinsic nuclease activity or recruit an as yet unidentified nuclease. Furthermore, northern blotting revealed that the processing or removal of some specific sncRNAs, *sncATP8/6* and *sncCO1*, appears to be perturbed in cells lacking *FASTKD4*, suggesting that the FASTKD proteins may also be involved in ncRNA clearance. Furthermore, the FASTKD proteins can have opposing functions in RNA processing. This is evident with the proximal

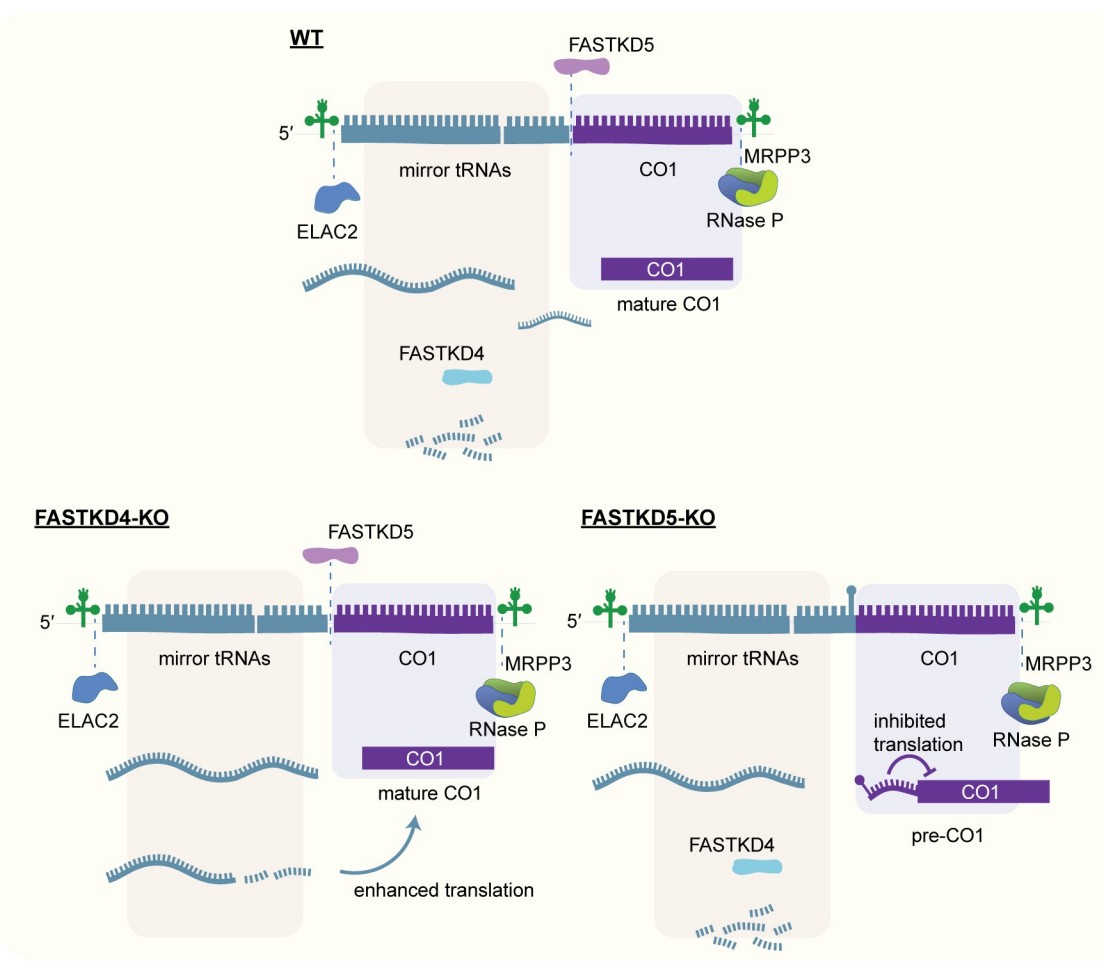

**Fig 7. A model showing the proposed roles of FASTKD4 and FASTKD5 in the RNA processing of the 5′ UTR of *CO1*.** In the absence of *FASTKD5*, the proximal processing site of the 5′ UTR of *CO1* cannot be resolved and obstructs mitoribosome binding, thereby precluding its translation. On the other hand, FASTKD4 facilitates the degradation of a cluster of mirror tRNAs (mirror $tRNA^Y$, $tRNA^C$, $tRNA^N$ and $tRNA^A$) that could play a regulatory role in COX1 translation.

processing site of the 5' UTR of *CO1* that is impaired in the absence of *FASTKD5*, while this junction is efficiently processed in cells lacking *FASTKD4*. In the absence of *FASTKD5*, the precursor *CO1* transcript containing the non-coding mirror $tRNA^Y$ is specifically increased, indicating that FASTKD5 facilitates the cleavage of the junction between mirror $tRNA^Y$ and the *CO1* mRNA. These findings are consistent with the significant reduction in COX1 protein synthesis in the loss of *FASTKD5*, and the increase of COX1 translation in the absence of *FASTKD4*. The underlying cause could be the presence of a structured stem-loop at the 5′ end of *CO1* that is not processed in the absence of *FASTKD5*, which obstructs mitoribosome recruitment and thereby inhibits its translation (Fig 7).

We have shown that several ncRNAs are differentially regulated in the *FASTKDs*-KO cell lines, and curiously, our analyses reveal that like the coding transcripts, ncRNAs are clearly processed and in most cases their lengths are similar to the length of the complementary mRNA. This suggests that both strands may be cleaved simultaneously in the same reaction. To explain this observation, we propose two contrasting hypotheses. First, that the initial substrate for processing could be a dsRNA comprising the two long complementary polycistronic RNAs, and that only after initial cleavage of the dsRNA, would smaller fragments be separated

by RNA helicases and subject to further maturation or degradation steps. Second, this could occur because antisense RNA sequences often fold into similar structures as their sense counterparts, such that "mirror" tRNAs and other complementary processing signals could be recognized by mitochondrial processing enzymes, as we have observed before for the RNase P complex [49].

A striking result from the HITS-CLIP analysis revealed that FASTKD4 preferentially binds to mitochondrial $tRNA^E$, which is complementary to the *ND5-CYTB* RNA processing site. Binding of FASTKD4 could prevent $tRNA^E$ from forming a duplex with its complementary region, which would have an inhibitory effect on *ND5-CYTB* RNA processing. If binding to RNA occurs prior to the processing of $tRNA^E$, FASTKD4 may bind to the $tRNA^E$ precursor transcripts containing *ND6* and its long 3′ UTR and may regulate the processing of other light strand transcripts encoded in the 3′ UTR of *ND6* including *ncND5*, *ncND4L/4*, *ncND3*, *ncCO3*, *ncATP8/6*, and *ncCO2*. Alternatively, the binding of FASTKD4 to $tRNA^E$ may reflect the possibility that this protein could, in addition to RNA processing, be involved in translation, given that a mitoribosomal protein MRPS27 also preferentially binds $tRNA^E$ as we have previously shown [53].

In conclusion, we have shown that FASTKD3, FASTKD4 and FASTKD5 regulate the expression of a broad range of mitochondrial transcripts, including ncRNAs, through controlling the processing and clearance of specific canonical and non-canonical junctions. We propose a new model, shown in Fig 7, whereby the orchestrated roles of FASTKD4 and FASTKD5 are crucial for the processing of the non-canonical pre-*CO1* mRNA in mitochondria by FASTKD5-mediated cleavage and removal of non-coding RNAs from that region by FASTKD4 that can regulate the translational efficiency of *CO1*.

## Materials and methods

### Cell culture

143B cells were cultured in Dulbecco's modified Eagle medium (DMEM) supplemented with 10% heat-inactivated fetal bovine serum (FBS), 2 mM L-glutamine, 100 U/ml penicillin, and 100 mg/ml streptomycin in 5% $CO_2$ at 37˚C. HeLa cells were cultured in DMEM supplemented with 10% FBS in 5% $CO_2$ at 37˚C. HAP1 cells were cultured in Iscove's Modified Dulbecco's Medium (IMDM) supplemented with 4 mM L-glutamine, 25 mM HEPES, 3.024 g/L NaHCO3, 10% heat-inactivated fetal bovine serum (FBS), 100 U/ml penicillin, 100 mg/ml streptomycin, 1 mM sodium pyruvate, and 50 μg/ml uridine in 5% $CO_2$ at 37˚C.

### Generation of CRISPR/Cas9 knockout cells

To generate *FASTKD*s-KO cells, we used Multiplex CRISPR/Cas9 Assembly Systems (Addgene Kit #1000000055) [54]. A single gRNA was cloned into pX330A or pX330S, and three gRNAs (*FASTKD3*, *FASTKD4*, and *FASTKD5*) were cloned into pX330A-1x3 following to the inventor's protocol [54]. Co-transfection of HAP1 cells with the appropriate gRNA constructs together with a GFP expression plasmid (pCI-GFP) was performed using Lipofectamine LTX (Invitrogen) according to the manufacturer's instructions. 24 h after transfection, GFP positive cells were selected by fluorescence activated cell sorting (FACS) using the Beckman Coulter MoFlo Astrios, and single cells were plated into individual wells of a 96-well plate. Screening of clones was performed by Sanger sequencing and immunoblot analyses.

### Northern blotting

Northern blot analyses were performed as previously described [55]. Briefly, 10 μg of total cellular RNA prepared using the TRIzol reagent was separated on denaturing formaldehyde 1%

agarose gel for mRNAs and rRNAs, or 10% polyacrylamide gel containing 7 M urea for tRNAs. RNAs were transferred to Hybond-N+ hybridization membranes (Sigma) and UV-crosslinked to the membrane. Membranes were hybridized with T7-transcribed [α-$^{32}$P] UTP radiolabeled riboprobes. Hybridization was carried out overnight at 65˚C in 50% formamide, 7% SDS, 0.2 M NaCl, 80 mM sodium phosphate (pH 7.4), and 100 mg/ml salmon sperm DNA. Imaging and quantification were performed using the Typhoon imaging system (GE Healthcare). Primers used for the preparation of PCR templates for the transcription of the riboprobes are listed in S1 Table. Northern blotting of the 5' UTR of the *CO1* mRNA was performed as previously described [12,13]. The following biotinylated oligonucleotide probes were used: /52-Bio/ (5' dual biotin modification)- atggctgagtgaagcattggac (antisense *tRNA$^Y$*), ggagaagatggttaggtctacg (*CO1* mRNA), and cctacggaaaccttgttacgac (*18S* rRNA).

## Measurement of respiratory chain activity

Measurement of oxygen consumption was performed using a Seahorse XFe24 Flux Analyzer (Seahorse Biosciences). 60,000 HAP1 cells were seeded into XF24 cell culture microplates coated with 0.01% poly-L-lysine solution (Sigma) and grown overnight. Cells were incubated at 37˚C for 1 h in Dulbecco's PBS with MgCl$_2$ and CaCl$_2$ (Sigma D8662) supplemented with 5 mM sodium pyruvate. Basal oxygen consumption was measured before further treatment. At the times indicated, the following compounds were injected: oligomycin (final concentration 2 μM), FCCP (final concentration 1 μM), rotenone and antimycin A (final concentration 1 μM each). Each measurement loop consisted of 30 s mixing, 1 min waiting, and 2 min measuring oxygen consumption. After the measurement, the number of cells was counted for normalization.

## Preparation of whole cell lysates and mitochondria-enriched fractions

Whole cell lysates (WCLs) were prepared by lysing cells in RIPA lysis buffer (50 mM Tris pH 8.0, 150 mM NaCl, 1% Triton X-100, 0.5% sodium deoxycholate, 0.1% SDS, and an in-house protease inhibitor cocktail) for 30 min on ice followed by centrifugation at 16,000 *g* for 10 min at 4˚C in order to remove insoluble material. For preparation of the mitochondria-enriched fraction, cell fractionation was carried out on ice in MB buffer (10 mM HEPES pH 7.4, 210 mM mannitol, 70 mM sucrose, 1 mM EDTA, and the in-house protease inhibitor cocktail) following 30 passages through a 25G x 1" Microlance needle (BD, 300400). Nuclei and unbroken cells were discarded after centrifugation at 2,000 *g*. The mitochondria-enriched fraction was then recovered by centrifugation at 10,000 *g*. Protein content was determined using the BCA Protein Assay Kit (Pierce).

## Immunoblotting and dot blotting

Equal amounts of protein were analyzed by SDS-PAGE. For Immunoblotting, proteins were transferred electrophoretically to nitrocellulose or PVDF membranes (GE Healthcare) and exposed to the primary antibodies specified below. The blots were further incubated with anti-goat, anti-rabbit or anti-mouse HRP-conjugated secondary antibodies (Dako) as appropriate, and visualized using ECL. For dot blots, the mitochondria-enriched fractions were lysed on ice for 30 min in 50 mM Tris pH 7.4, 150 mM NaCl, 5 mM EDTA, 0.5% NP40, in-house protease inhibitor cocktail, and 1U/μL RNasin Plus RNase Inhibitor (Promega). Soluble material was collected following centrifugation for 30 min at 21,000 *g*. Equal amount of protein were spotted onto a nitrocellulose membrane (GE Healthcare). The membrane was exposed to antibodies and signals were visualized by chemiluminescence. The following primary antibodies were used: anti-FASTK (Abcam, ab97544), anti-FASTKD2 (Proteintech, 17464-1-AP), anti-

FASTKD3 (Sigma-Aldrich, sab2701102), anti-FASTKD4 (Proteintech, 16245-1-AP), anti-FASTKD5 (Sigma-Aldrich, sab2700438), anti-dsRNA (J2; Scicons, 10010200), anti-mtHSP70 (Invitrogen, MA3-028), anti-VDAC1 (Santa Cruz, sc-8829), anti-γ tubulin (Santa Cruz, sc-17787), anti-GRSF1 (Atlas antibodies, HPA036985), anti-LRPPRC (Santa Cruz, sc-166178), anti-PNPT1 (Santa Cruz, sc-365049), anti-DHX30 (Abcam, ab85687), anti-RPUSD4 (Abcam, ab122571), anti-SUV3 (Santa Cruz, sc-365750).

## Immunofluorescence

Immunofluorescence analyses were performed on cells fixed in 4% paraformaldehyde for 15 min. Cell permeabilization and blocking were performed together by incubating the fixed cells in PBS containing 0.3% Triton X-100 and 1% pre-immune goat serum for 45 min. The same buffer was used to incubate cells with the specified primary antibody. After 90 min incubation, the cells were washed in PBS and incubated with the appropriate secondary antibody conjugated with AlexaFluor 488 or Alexa 594 (Life Technologies). Mitochondrial networks and nascent transcripts were stained respectively using MitoTracker Red FM (Thermo Fisher Scientific), and 5 mM BrU (Sigma) as previously described [16]. Imaging was performed using a Zeiss LSM700 confocal microscope equipped with a 63× oil objective. All images were imported into ImageJ and uniformly adjusted for brightness and contrast. The following primary antibodies were used: anti-SSBP1 (Proteintech, 12212-1-AP), anti-BrU (Roche, 11170376001), anti-dsRNA (J2; Scicons, 10010200).

## RNA-seq

RNA sequencing was performed on total RNA from three control and three of each of the five *FASTKD* knockout cell lines, using the Illumina NovaSeq platform, according to the Illumina Tru-Seq protocol as described previously [12,13,29]. Raw reads were trimmed of adapter sequences using cutadapt v1.18 [56] with standard parameters. Trimmed reads were mapped against the NuMTS masked human genome sequence (Hg38) using STAR v2.7.3a [57] and GENCODE v34 gene annotation with a customized mitochondrial annotation. Salmon v1.2.1 [58] (-l ISR—seqBias—-gcBias) in alignment based mode was used to quantify gene expression from the transcriptome alignments produced by STAR with a sequence file produced by gffread v0.11.7 [59]. Strand-specific coverage of the primary alignments of properly paired, full-length RNA fragments (excluding those with a length > 500 nt) across the mitochondrial genome was calculated with samtools v1.10 [60] and bedtools genomecov v2.26.0 [61], normalised to library size. Differential expression analysis was performed with DESeq2 by counts summarized with tximport [62].

## Poly(A) analyses

Sequenced reads were aligned against the human mitochondrial transcriptome bowtie2 v2.4.1 (—nofw—fr) [63] using the GENCODE v34 transcript set combined with custom mitochondrial rRNA and mRNA sequences with an additional 50 adenine residues added to the 3′ ends to allow mapping of polyA tails as we have described previously [64]. Properly paired alignments normalized to library size were calculated with samtools v1.10 [60] and bedtools genomecov v2.26.0 [61]. Gene-level normalized coverage profiles were extracted and lengths of poly(A) tails analyzed [61,64].

## BioID

Using the GAL4/UAS expression system, 4-hydroxytamoxifen (4-OHT) inducible 143B cells were generated for temporal control of protein expression of FASTKD4 or FASTKD5 tagged

with BirA (R118G)-Flag at the C-terminus [65]. Briefly, the *FASTKD4* or *FASTKD5* coding sequence tagged with C-terminal BirA (R118G)-Flag was cloned into pF 5xUAS plasmid (a kind gift from Prof. John Silke, The Walter and Eliza Hall Institute of Medical Research) [65]. Plasmids were packaged into lentivirus and used to infect 143B cells containing a genomic GAL4-ER$^{T2}$-VP16 cassette. Cells were selected using 3 mg/ml puromycin. Protein expression was induced following addition of 100 nM 4-OHT for 24 h and biotinylation was then induced by addition of 50 μM biotin for 1 h. Mitochondria were extracted from cells to minimize contamination with free biotin, and subjected to pulldown experiments following to the previously published protocol [66] with the following minor modifications: mitochondria were lysed on ice for 30 min without sonication; and proteins were eluted in 1x Laemmli buffer by heating at 95˚C for 10 min and further analyzed by LC-MS/MS.

### LC-MS/MS

Peptides were desalted on C18 StageTips [67] and dried by vacuum centrifugation prior to LC-MS/MS injections. Samples were resuspended in 2% acetonitrile, 0.1% FA and nano-flow separations were performed on a Dionex Ultimate 3000 RSLC nano UPLC system connected in-line with a Lumos Fusion Orbitrap Mass Spectrometer. A capillary precolumn (Acclaim Pepmap C18, 3 μm-100 Å, 2 cm x 75 μm ID) was used for sample trapping and cleaning. A 50 cm long capillary column (75 μm ID; in-house packed using ReproSil-Pur C18-AQ 1.9 μm silica beads; Dr. Maisch) was then used for analytical separations at 250 nl/min over 150 min biphasic gradients. Acquisitions were performed through Top Speed Data-Dependent acquisition mode using a cycle time of 1 s. Initial MS scans were acquired with a resolution of 240'000 (at 200 m/z) and the most intense parent ions were selected and fragmented by High energy Collision Dissociation (HCD) with a Normalized Collision Energy [68] of 30% using an isolation window of 0.7 m/z. Fragmented ions were acquired using the Ion Trap using a maximum injection time of 20 s. Selected ions were then excluded for the following 20 s. Raw data were processed using SEQUEST, MS Amanda and Mascot in Proteome Discoverer v.2.4 against a concatenated database consisting of the Uniprot Human protein database (73112 entries) and a FASTKD sequence. Enzyme specificity was set to trypsin and a minimum of six amino acids was required for peptide identification. Up to two missed cleavages were allowed. A 1% FDR cut-off was applied at both peptide and protein identification levels. For the database search, carbamidomethylation was set as a fixed modification, whereas oxidation (M), acetylation (protein N-term), PyroGlu (N-term Q), and Phosphorylation (S,T,Y) were considered as variable modifications. Data was further processed and inspected in Scaffold 4.10 (Proteome Software, Portland, USA) and spectra of interest were manually validated.

### High-throughput sequencing of RNA isolated by cross-linking immunoprecipitation (HITS-CLIP)

Covalent bonds between endogenous FASTKD4 protein and RNA in HeLa cells were induced by irradiation with ultraviolet (UV) light (150 mJ/cm$^2$). After cell collection, mitochondria were extracted by differential centrifugation. Pelleted mitochondria were lysed and the samples were incubated overnight with 5 μg anti-FASTKD4 (Proteintech, 16245-1-AP) or anti-HA (Roche, 11867423001) antibody as a control, in the presence of proteinase and ribonuclease inhibitors. Protein-antibody complexes were then bound to Pierce Protein G Magnetic Beads (Thermo) and while still on beads, RNA was partially digested with micrococcal nuclease (NEB) for 10 min at room temperature, in the presence of TURBO DNase. The reaction was stopped by adding 12 μl SUPERasin (ThermoFisher Scientific). After washing and elution of proteins from the beads, proteins were digested with 4 mg/ml Proteinase K. RNA was end

repaired with Antarctic Phosphatase and T4 Polynucleotide Kinase and used for library generation, following the manufacturer's protocol (TruSeq Small RNA library preparation kit (Illumina)). Indexed PCR product was run on a 6% PAGE TBE gel to remove adapter only fragments and purified with Zymo DNA Clean and Concentrator columns. Quality and concentration were assessed with a High Sensitivity D1000 Screentape for TapeStation (Agilent). Libraries were subjected to high-throughput sequencing using the Illumina HiSeq platform. For the qPCR analysis, the protein-RNA complexes were similarly immunoprecipitated from WT and *FASTKD4*-sKO HAP1 cells, then the RNA was recovered without digestion. Anti-Flag (Sigma, F7425) antibody was used as control.

## Processing and mapping of HITS-CLIP data

Quality trimming and 3′-end adaptor clipping of sequenced reads were performed with Trim Galore!. Sequenced reads longer than 20 nt were aligned to the human reference genome (GRCh38) with Bowtie2 [63]. The alignment was split by template strand (Samtools) [60] and converted to BED files (Bedtools) [61], and peak calling was performed with Piranha [69].

## Pulse labeling of mitochondrial translation

Pulse labeling experiments to evaluate mitochondrial translation were carried out as described previously [55]. Briefly, HAP1 cells were incubated for 20 min in methionine and cysteine-free DMEM supplemented with 10% heat-inactivated FBS, 110 mg/L sodium pyruvate, and 2 mM L-glutamine. 100 μg/ml emetine dihydrochloride was added for 5 min to inhibit cytosolic translation, followed by addition of 200 μCi/ml $^{35}$S-labeled methionine and cysteine mix (Perkin Elmer). Labelling was performed for 1 h, then cells were lysed. Equal amounts of each protein sample were resolved on 12–20% SDS-PAGE gels. Gels were stained with Coomassie Brilliant Blue to confirm equal loading, then dried and exposed. Imaging and quantification were performed with the Typhoon imaging system (GE Healthcare).

## Quantitative PCR (qPCR) and Rapid amplification of cDNA ends (RACE)

Total cellular DNA purified by phenol-chloroform extraction was analyzed using GoTaq qPCR Master Mix (Promega) following the manufacturer's instructions. For RT-qPCR, cDNA was prepared from DNase-treated RNA using the GoScript Reverse Transcriptase (Promega) with random primers (Promega, C118A) or RNA-specific primers for strand-specific qPCR. Primers used for RT and qPCR are listed in S1 Table. RACE was carried out as previously described in [49].

## *C. elegans* strain generation and culture

*C. elegans* strains were maintained on OP50 plates using standard conditions at 20˚C. N2 (Bristol strain) was used as a wild type, and the deletion allele *fask-1(uge100)* (strain FAS150) was derived from this background, using CRISPR/Cas9 as described [70]. The deletion allele is described in S8A Fig. The sgRNAs targeted the following genomic sequences: atatctagaatgtg-gagccgt<u>gg</u>, <u>ccc</u>acgctttgcaggacgattcc, tttcagaacacagaaacgagc<u>gg</u>, and <u>ccg</u>ctggcttcgatccagtagtc (PAM sequences are underlined). The deletion allele was identified by single worm PCR and sequenced with the primers oFSa0222 (agtcatggatgcagcgaaag) and oFSa0223 (gtggtgtca-gagtgtctcat). For RNA analysis, worms from eight 9-cm nematode growth medium plates containing mostly adult, embryo and young larval stages were washed three times with PBS. Worms were pelleted after each wash by centrifugation for 2 min at 1000 *g*, and the washed worm pellets were resuspended in 2 ml Trizol reagent and frozen at -80˚C.

## Supporting information

**S1 Fig. Sequence analysis of the *FASTKDs*-KO genes in the HAP1 cell lines.** Genomic modifications shown in (**A**) were detailed in (**B**).
(TIF)

**S2 Fig. RNA-seq of the *FASTKDs*-KO genes in the HAP1 cell lines.** Specific deletions in each of the targeted *FASTKD* genes in each of the HAP1 cell lines relative to the control HAP1 cell line were shown.
(TIF)

**S3 Fig. Characterization of the *FASTKDs*-KO HAP1 cell lines.** (**A**) Protein expression levels in 143B (control) and WT HAP1 cells were assessed by immunoblotting. Whole cell lysates (WCL) and mitochondria extracted from the cells were analyzed. mtHSP70 was used as a loading control. (**B**) mRNA expression levels in 143B (control) and WT HAP1 cells were analyzed by RT-PCR. (**C**) Relative abundance of *7S* RNA in WT (control) and *FASTKDs*-KO HAP1 cells were quantified by RT-qPCR (n = 4). The amounts of *7S* RNA were normalized by *GAPDH*. (**D**) The relative abundance of mtDNA in the *FASTKDs*-KO cells compared to controls was measured by qPCR and normalized to total nuclear DNA (n = 3). (**E**) WT and *FASTKDs*-KO HAP1 cells were labeled with BrU, MitoTracker Red, and DAPI, and immuno-labeled with anti-SSBP1, anti-BrU and J2 anti-dsRNA. Immunofluorensce was analyzed by confocal microscopy. Scale bars indicate 20 μm. (**F**) The amount of mitochondrial dsRNA in the *FASTKDs*-KO cells and controls was measured by dot blotting. VDAC1 protein was used as a loading control. (**G**) Steady-state levels of mitochondrial tRNAs in WT (control) and *FASTKDs*-KO HAP1 cells were measured by northern blotting. *5S* was analyzed as a loading control.
(TIF)

**S4 Fig. The steady-state levels of rRNAs, mRNAs and ncRNAs in WT and *FASTKDs*-KO HAP1 cells measured by northern blotting. Related to Fig 2.** Two entire gels/exposures are shown. 5S and 7SL were used as loading controls. Equal amounts of total cellular RNA were loaded on the gel. Independent biological samples were analyzed per genotype. The pair of WT #2 and *FASTKD4*-sKO (A), a pair of WT #2 and *FASTKD3&4*-dKO (A), a pair of WT #1 and *FASTKD4&5*-dKO (B), and a pair of WT #3 and *FASTKD3*-sKO (B) are also shown in Fig 2A.
(TIF)

**S5 Fig. Rapid amplification of cDNA ends (RACE) and northern blotting of the 5′ UTR of the *CO1* mRNA.** (**A**) RACE of the 5′ UTR of *CO1* mRNA in WT (control) and *FASTKDs*-KO HAP1 cells. (**B**) Sanger sequencing of the 5′ UTR of *CO1* mRNA from WT HAP1 cells. (**C**) RACE of the 5′ end of *CO2* mRNA in WT (control) and *FASTKDs*-KO HAP1 cells. (**D**) Northern blotting of RNA isolated from WT (control) and *FASTKDs*-KO HAP1 cells against the antisense *tRNA*^Y, *CO1* mRNA, and *18S* rRNA that was used as a loading control.
(TIFF)

**S6 Fig. Polyadenylation status of mitochondrial mRNAs. Related to Fig 4.** Mapping of RNA-seq reads to polyadenylated mitochondrial mRNA reference sequences was used to analyze the polyadenylation status of mRNAs in the *FASTKDs*-KO cell lines compared to controls. The canonical 3′ region of each mRNA is shown in dark grey and the section of polyadenosine is shown in light grey.
(TIF)

**S7 Fig. FASTKD4 specifically binds to *tRNA$^E$*. Related to Fig 5.** (**A-B**) Strand specific HIT-S-CLIP read counts corresponding to the 13401–16569 region of the mtDNA for control (anti-HA antibody) (A) and anti-FASTKD4 antibody (B). (**C**) The *tRNA$^E$* enriched through FASTKD4 immunoprecipitation from WT and *FASTKD4*-sKO HAP1 cells was quantified by strand-specific RT-qPCR. Anti-Flag antibody was used a s control. A representative result from two independent biological experiments is shown.
(TIF)

**S8 Fig. BioID interactome analyses of FASTKD4 and FASTKD5.** Log$_2$FC (FASTKD4-BirA/control) (**A**) or Log$_2$FC (FASTKD5-BirA/control) (**B**) was plotted against a sum of normalized total spectra. Only significantly enriched proteins are shown (Log$_2$FC $\geq$ 1). Cells that do not express the bait protein were used as control. Mitochondrial proteins are shown in blue. Red dot represents BirA-tagged bait proteins. (**C-D**) Gene ontology analysis (GO—molecular function) of mitochondrial proteins identified by the BioID of FASTKD4 (C) and FASTKD5 (D). (**E**) Number of mitochondrial proteins found in either or both BioID data sets from FASTKD4 or FASTKD5.
(TIF)

**S9 Fig. Function of the FASTK family proteins in the processing of tRNA-less junctions is evolutionarily conserved.** (**A**) A schematic representation of *fask-1 (B0564.7)* deletion allele generated using CRISPR/Cas9 technology. Two sgRNAs each were used to cleave the genomic DNA near the start codon and in the last exon of the FASK-1 locus. The deletion allele *fask-1 (uge100)* contains a truncated start codon, a 26 bp insertion and a 3642 bp deletion that removes all of the FASK-1 coding sequence except 153 bp of the last exon. (**B**) The steady-state levels of mitochondrial transcripts in WT (control) and *FASK-1* deletion strains were measured by northern blotting. *Act-1* was used as a loading control.
(TIF)

**S1 Table. List of oligonucleotides used in this study.** Sequences used for Gibson assembly are shown in bold. T7 promoter sequence is underlined.
(TIF)

## Acknowledgments

We would like to thank all current and past members of the JCM lab for their helpful advice and intellectual input during the course of this work. We would also like to thank Prof. John Silke (The Walter and Eliza Hall Institute of Medical Research) for sharing plasmids, and Prof. Walter Rossmanith (Medical University of Vienna) and Kinsey Maundrell (University of Geneva) for helpful discussions about our results. We would additionally like to thank Romain Hamelin and Diego Chiappe (Proteomics Core Facility, EPFL) who carried out the LC-MS/MS.

## Author Contributions

**Conceptualization:** Aleksandra Filipovska, Jean-Claude Martinou.

**Data curation:** Akira Ohkubo, Lindsey Van Haute, Danielle L. Rudler, Maike Stentenbach.

**Formal analysis:** Akira Ohkubo, Lindsey Van Haute, Danielle L. Rudler, Maike Stentenbach, Oliver Rackham, Aleksandra Filipovska.

**Funding acquisition:** Florian A. Steiner, Oliver Rackham, Michal Minczuk, Aleksandra Filipovska, Jean-Claude Martinou.

**Investigation:** Akira Ohkubo, Lindsey Van Haute, Danielle L. Rudler, Maike Stentenbach, Florian A. Steiner.

**Methodology:** Akira Ohkubo, Lindsey Van Haute, Danielle L. Rudler, Maike Stentenbach, Florian A. Steiner.

**Project administration:** Akira Ohkubo.

**Resources:** Florian A. Steiner, Oliver Rackham, Michal Minczuk, Aleksandra Filipovska, Jean-Claude Martinou.

**Supervision:** Florian A. Steiner, Oliver Rackham, Michal Minczuk, Aleksandra Filipovska, Jean-Claude Martinou.

**Visualization:** Akira Ohkubo, Lindsey Van Haute, Danielle L. Rudler, Florian A. Steiner.

**Writing – original draft:** Akira Ohkubo, Lindsey Van Haute, Florian A. Steiner, Oliver Rackham, Aleksandra Filipovska.

**Writing – review & editing:** Aleksandra Filipovska, Jean-Claude Martinou.

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
