## [Decision Letter · Decision Letter 0]

7 Jun 2021

Dear Dr Ohkubo,

Thank you very much for submitting your Research Article entitled 'The FASTK family proteins fine-tune mitochondrial RNA processing' to PLOS Genetics.

The manuscript was fully evaluated at the editorial level and by independent peer reviewers. The reviewers appreciated the attention to an important problem, but raised some substantial concerns about the current manuscript. Based on the reviews, we will not be able to accept this version of the manuscript, but we would be willing to review a much-revised version. We cannot, of course, promise publication at that time.

If you decide to revise the manuscript for further consideration at PLOS Genetics, please aim to resubmit within the next 60 days, unless it will take extra time to address the concerns of the reviewers, in which case we would appreciate an expected resubmission date by email to plosgenetics@plos.org.

[LINK]

We are sorry that we cannot be more positive about your manuscript at this stage. Please do not hesitate to contact us if you have any concerns or questions.

Yours sincerely,

Christoph Freyer, PhD

Guest Editor

PLOS Genetics

Gregory Barsh

Editor-in-Chief

PLOS Genetics

Guest editor comments:

· Given the haploid state of the cell line used, I would appreciate if you could clarify why the identified FASTKD5 variant in the FASTKD3/4-dKO cells is deemed as irrelevant to the experiments. The Western blot in Figure 1 would suggest otherwise.

· The conclusions of the Northern blot experiments in figure 2 seem to be based on statistical significance, rather than physiological relevance. Some of the signals are very low, making quantification difficult, increasing variation and therefore potentially skewing the conclusions. The lack of error bars for WT samples in figures 2B-E, makes interpretation more difficult. This is especially true for some of the nc and sncRNA species. It woudl be helpful in some cases, such as for ND6 or ncND3, to indicate what parts of the blot were used for quantification.

· Please ensure the readers in the figure legend that the presented gel sections in figure 2A stem from the same representative gel and not from different gels. It might also be useful to provide one set of images in the supplementary files.

· I recommend checking the manuscript for inconsistencies. For instance, FASTKD5 levels in FASTKD3&4-dKO (line 140) and FASTKD3-sKO cells (line 157). Interaction between FASTKD3 and FASTKD5 (line 159), but not in BioID data.

· It would help the reader to clarify that the BioID was performed in a polyploid cell line, which expresses FASTKD2, unlike the haploid cell line used throughout the manuscript. The reader might otherwise wonder how an interaction with in the FASTKD4-BirA BioID experiment was possible (Fig. S7).

· Clarification of the RNAseq data and polyadenylation experiment will benefit clarity of the results.

· In general, the manuscript could benefit from clarity and less speculation in the results section.

· It would help the reader if the canonical and non-canonical transcripts are defined and the transcript boarders described.

Reviewer's Responses to Questions

**Comments to the Authors:**

Reviewer #1: The manuscript describes the results of establishing cell lines in which the expression of one or two of the mitochondrial FASTKD proteins is knocked down. These proteins are involved in the RNA processing and post-transcriptional regulation of gene expression in the mitochondria. The work in this manuscript attempted looking and unraveling the function of each one of these family members in a comprehensive approach. The question of the control of gene expression of mitochondrial encoded genes is an important one and under an extensive study. Therefore, this manuscript would interest those who study mitochondrial gene expression, as well as those interested in RNA processing and stability. Although a defined clear picture of the mechanistic role of each member is still not at hand, this work is an important step towards this goal.

Major comments:

The authors use the terms “canonical” versus “non-canonical” transcripts throughout the manuscript. However, it is not clearly defined what is each one. Is a canonical transcript is the mature processed one or a transcript that is composed of the amino-acids coding region (and a polyA tail). For example, the matured processed CO1 has been mapped to have the tRNAS antisense serving as its 3’ UTR. Similarly, the mature ND6 transcript has been mapped to have a long 3’ UTR of about 450 nt (Slomovic et al MCB 2005). Therefore, all the mitochondrial transcripts should be precisely defined and described as to their 5’ and 3’ ends, even if they are surrounded by tRNAs. This is also applied for the polyadenylation sites (see below).

For example, the term pre-CO1 is used but no description of what exactly is that RNA molecule. The presented RACE results are of low resolution gel and the nt sequences of the RACE clones are not shown.

The same account to Fig 7 where the RNA molecules are schematically presented but the reader cannot identify the exact transcripts. These should be indicated using the mit genomic nt sequence numbering or a similar mapping system.

Fig 4 and S5 Fig. It is not clear what is presented here. Is the canonical mRNA region ends at the translational stop codon or at the mature 3’ end of the transcript? As said, these are not the same for several transcripts. Is the section of polyadenylation representing a polyA sequence or the 3’ UTR of the RNA / the seq 3’ to the CDS? This is not explained in the legends to figures and the ref indicated does not seems to be the right one. In addition, the title of Fig 4 seems to be not the right one and in the legend to S5 Fig it is indicated “reveals effects on polyadenylation in all mRNAs” which is not the case. For example, the last column in S5 Fig present the ND6 mRNA polyadenylation. This mature transcript is about 950 nt long and the polyadenylation is at the 3’ end. Only nt 445 to 525 sequence is presented here so where is the mature 3’ end?

Minor:

1. S3E Fig. The Bru and dsRNA staining is very faint and hard to detect.

2. What is the ND5-cytB junction? Define it clearly.

3. S4C Fig. The blot of the pre-CO1 is too faint.

4. Ref 64 does not seem the right one for the polyadenylation method.

Reviewer #2: Non canonical is given as “mitochondria, junctions not flanked by tRNA are found at the 3′ UTR of ND6, 5′ UTR of CO1, between ND5 and CYTB, and between ATP8/6 and CO3 mRNAs. “ This is a little misleading as some of these junctions do have tRNA like structures as they are derived from the antisense of the tRNA and are predicted to fold into a very similar structure. Thus the enzyme that facilitate the cleavage may be same as for junctions between ORFs and sense strand mt-tRNAs.

Fig 1 Although a number of probes have been shown at 2 exposure times, a number of the signals are overexposed (eg GRSF1, VDAC) and so any more subtle differences in levels are lost. The quality of some of the probe exposures is rather poor eg FASTKD3.

Fig 2 This is an extensive display of data but unfortunately not acceptable as essentially all of the signals are cut and paste of a single band. As a mosaic it is therefore impossible to know what was on the same blot, whether the loading was equal or represents the best from a composite of many individual blots. There are also no size markers to validate the identity of the signals.

The text states that “all the analyzed antisense or ncRNAs appeared as distinct bands with precise lengths”. There are no size markers provided so it is not possible to determine is the largest species is the correct length, non-specific larger species, unprocessed, or a discrete degradation product. This is particularly important considering in most of the antisense panels there are more than one species shown.

In this same section it states that “length of some ncRNAs is similar to the length of their complementary mRNAs,”. Without any size markers it is not really possible to make this conclusion.

It is therefore difficult to fully assess the validity of the data presented in this figure.

The lack of agreement, acknowledged by the authors, between the RNAseq and the northerns for some of the transcripts was a little concerning, as it makes it more difficult to have confidence in the remaining RNAseq data.

Fig 3B reflects a large amount of data. FASTKD5 KO in 3B is described as “ resulted in the accumulation of transcripts or junction containing RNAs that are antisense to the ND2, CO2, ATP8/6, CO3 and ND6 mRNAs,..”. If this refers to the red in the LSP derived trace, then it is not the whole RNA species that has elevated levels in a couple of the cases and an increase (red above the line) is not really evident for NT6. If this is not what is meant then the text is not clear to this reader.

Fig 4 – the term non-template adenylation is not used in the main text that uses polyadenylation. A degree of consistency in terminology would be helpful. The change in adenylation status is not clearly described or depicted in the figure. To the less knowledgeable reader the nucleotide position might be assumed to be the nucleotide position in the mtDNA whereas it reflects the length of the mature transcript. Considering the manuscript is about non-correctly processed it would be better to relate the nucleotide position to the mtDNA position. It would also be helpful to have an indicator of the correct end following processing as this will help to align where the polyadenylation might be expected to start. The change in colour to designate this is not made clear in the legend or the text. What is also not clear is what the drop off in the graph is meant to indicate. Does it mean that FASTKD3 KO MTCO2has a population of transcripts with shorter A tails ? Has this been validated by other means. If this is not what it signifies then it is not clear to this reader. This is particularly important when concluding that there was a difference in the FASTKD4 and FASTKD5 KOs, because this is difficult to see compared to traces for control and for the traces in the other graphs.

Minor points

Line 73 ‘span’ probably should be exchanged with ‘intersperse’ as span suggests coverage over the non-coding regions.

The lack of spacing between each transcript in Fig 3A makes it hard to take in the data. It is better presented in Fig 2.

Reviewer #3: The manuscript by Ohkubo and colleagues analyzed the consequences of deleting different FASTK genes in mitochondrial RNA processing and levels. Loss of FASTKD4 and FASTKD5 and their double knockout showed a strong respiratory phenotype, with decreases in CO1 and ND5. Comprehensive mitochondrial transcriptome analyses showed defects in processing at several canonical and non-canonical RNA junctions, accompanied by an increase in specific antisense transcripts. There was no effect on poly adenylation.

This comprehensive study allowed for a direct comparison between the FASTK forms and expanded our understanding of mitochondrial RNA processing mechanisms. I have relatively minor comments.

1- I was a bit confused with the cell line “with FASTKD3 and FASTKD4 genes were deleted also carried a mutation in one of the FASTKD5 alleles” HAP1 cells being haploid, the deletion of one FASTKD5 allele would be of consequence. Maybe I am missing something.

2- I was also confused with the western blots. FASTKD5 bands can still be observed in the single and double KOs. Please clarify.

3- The observation that “degradation. The fact that the length of some ncRNAs is similar to the length of their complementary mRNAs, which suggests that the RNA processing of the two strands may occur simultaneously” is difficult to judge, as the bands do not have MW markers and are in different panels.

4- If the translatability of ND5-CYTb transcript is correct, should we observe a decrease in cytochrome b protein? The gels shows an overlap with ND2, but please expand. The western blot on Figure 6 does show a decrease in UQCRC2, which may be due to lower cytochrome b.

5- The binding of FASTKD4 to tRNA E is indeed interesting, but it may just reflect the fact that it binds to the unprocessed transcript at that same sequence. Still, I agree that there should be preferential binding to the unprocessed.

**Have all data underlying the figures and results presented in the manuscript been provided?**

Reviewer #1: Yes

Reviewer #2: **No: **Please see comment on Fig 2

Reviewer #3: Yes

PLOS authors have the option to publish the peer review history of their article (what does this mean?). If published, this will include your full peer review and any attached files.

Reviewer #1: No

Reviewer #2: No

Reviewer #3: **Yes: **Carlos T Moraes

---

## [Decision Letter · Decision Letter 1]

28 Sep 2021

Dear Dr Ohkubo,

Thank you very much for submitting your Research Article entitled 'The FASTK family proteins fine-tune mitochondrial RNA processing' to PLOS Genetics.

The manuscript was evaluated by the original 2 peer reviewers. As you will see, reviewer 1 is positive but reviewer 2 remains concerned regarding Figure 2.

The manuscript and the reviews have now been discussed among members of the editorial board. Overall, we are positive about moving forward but ask that you make some additional changes to the presentation in a hopefully final round of minor revision. We recommend that, where possible, you provide images in figure 2 where WT and KO are loaded adjacent. You already provide this for FASTKD3-sKO, FASTKD4-sKO and you provided such an image to the reviewers (middle two lanes of Figure II in your response letter). Should you not have such configurations for the dKOs, please provide full images with WT and dKO on the same exposure as supplemental information (similar as you provided for reviewers only). All these images should be presented as supplemental data, as part of the revised manuscript. We also recommend to clearly state in the results and methods sections how the figure was composed and, if any, which lane contains a re-used WT control (i.e. it seems that the same WT blot is shown for FASTKD4-sKO and FASTKD3&4-dKO). Finally, we suggest that you clarify the polyA tail length in figure 4.

We hope to receive your revised manuscript within the next 30 days. If you anticipate any delay in its return, we would ask you to let us know the expected resubmission date by email to plosgenetics@plos.org. If present, accompanying reviewer attachments should be included with this email; please notify the journal office if any appear to be missing. They will also be available for download from the link below. You can use this link to log into the system when you are ready to submit a revised version, having first consulted our Submission Checklist.

[LINK]

Yours sincerely,

Christoph Freyer, PhD

Guest Editor

PLOS Genetics

Gregory Barsh

Editor-in-Chief

PLOS Genetics

Reviewer's Responses to Questions

**Comments to the Authors:**

Reviewer #1: The authors have answered the questions and changed the manuscript accordingly to the comments.

Reviewer #2: The authors have amended the publication mostly by eliminating text that was considered to be inappropriate by the reviewers.

Figure 2 is a crucial element of the manuscript containing data that is fundamental to the premise being presented. In response to comments about the fragmentary nature of Figure 2, there has been the addition of 2 sentences to state that the gel images in Fig 2 were pieces excised and from a single gel and re-assembled into a major figure. It also suggests that all the data were derived from a single gel/membrane. The example submitted to the editor/reviewers in the revision only shows FASTKD5 sKO suggesting that there were separate gels for each KO pairing, or where the same WT repeats used for each WT lane in Figure 2 ? The WT lanes would be expected to be identical, especially if the loading were equal and the same exposures retained, which they seem to be for the 5S. These do however differ between WT signals for many of the RNA species probed.

The additional figure for the reviewers is not at the same exposure/brightness as the manuscript version, so it is not completely possible to judge but is it the same pair of lanes used for each probe ? Since 5S and 7SL are given as loading controls and these appear to be reasonably equally loaded, why not use WT rep3 and FASTKD5 sKO rep1. That would allow these to be presented as a pair of signals in a single piece, as was done for FASTKD3 sKO and FASTKD4 sKO data, which would be marginally better than the original and current format, instead of 2 separate pieces.

WT NcND3 in fig 2 have no corresponding signal to that of the fastest mobility in the FASTKD5 sKO lane. This is different to the figure submitted to the reviewers as an example of the original gel, where the fast migrating doublet is visible in all 6 lanes.

This is not really acceptable quality for a journal such as PLOS Genetics. It is not evident that all of the sample were loaded on the same gel. The northern probings should be repeated with the samples loaded in the required order on a single gel that can be shown in its entirety.

These elements when taken together mean this figure is still below the quality this reviewer would expect of a PLOS Genetics manuscript.

Fig 4 seems to suggest that the poly(A) tail in WT controls is generally 15 nucleotides, which does not correspond with most of the literature including that cited by the authors in their response (Temperley et al 2010).

The comment about radioactive RNA markers is understood but RNA markers are available to run on the gels. Images of these on UV together with the 28S and 18S would allow for subsequent sizing to be shown.

**Have all data underlying the figures and results presented in the manuscript been provided?**

Reviewer #1: Yes

Reviewer #2: Yes

PLOS authors have the option to publish the peer review history of their article (what does this mean?). If published, this will include your full peer review and any attached files.

Reviewer #1: **Yes: **Gadi Schuster

Reviewer #2: No

---

## [Editor Report · Decision Letter 2]

11 Oct 2021

Dear Dr Ohkubo,

We are pleased to inform you that your manuscript entitled "The FASTK family proteins fine-tune mitochondrial RNA processing" has been editorially accepted for publication in PLOS Genetics. Congratulations!

Yours sincerely,

Christoph Freyer, PhD

Guest Editor

PLOS Genetics

Gregory Barsh

Editor-in-Chief

PLOS Genetics

Comments from the reviewers (if applicable):

**Data Deposition**

http://datadryad.org/submit?journalID=pgenetics&manu=PGENETICS-D-21-00596R2

**Press Queries**

---

## [Editor Report · Acceptance letter]

3 Nov 2021

PGENETICS-D-21-00596R2 

The FASTK family proteins fine-tune mitochondrial RNA processing 

Dear Dr Ohkubo, 

We are pleased to inform you that your manuscript entitled "The FASTK family proteins fine-tune mitochondrial RNA processing" has been formally accepted for publication in PLOS Genetics! Your manuscript is now with our production department and you will be notified of the publication date in due course.

With kind regards,

Andrea Szabo

PLOS Genetics

On behalf of:
